# Surging of a Hudson Strait-scale ice stream: subglacial hydrology matters but the process details mostly do not

**Matthew Drew and Lev Tarasov**

Department of Physics & Physical Oceanography, Memorial University of Newfoundland, St John's, Canada

**Correspondence:** Matthew Drew (mdrew@mun.ca)

**Abstract.** While subglacial hydrology is known to play a role in glacial dynamics on sub-annual to decadal scales, it remains unclear whether subglacial hydrology plays a critical role in ice sheet evolution on centennial or longer timescales. Furthermore, several drainage systems have been inferred, but it is unclear which is most applicable at the continental/glacial scale. More fundamentally, it is even unclear if the structural choice of subglacial hydrology truly matters for this context.

Here we compare the contribution to the surge behaviour of an idealized Hudson Strait-like ice stream from three subglacial hydrology systems. We use the newly updated BAsal Hydrology Model – BrAHMs2.0 – and provide model verification tests. BrAHMs2.0 incorporates two process-based representations of inefficient drainage dominant in the literature (linked cavity and poro-elastic) and a non-mass-conserving zero-dimensional form (herein termed leaky bucket) coupled to an ice sheet systems model (the Glacial Systems Model, GSM). The linked-cavity and poro-elastic configurations include an efficient drainage scheme while the leaky bucket does not. All three systems have a positive feedback on ice velocity, whereby faster basal velocities increase melt supply. The poro-elastic and leaky-bucket systems have diagnostic effective pressure relationships – only the linked-cavity system has an additional negative feedback, whereby faster basal ice velocities increase the dynamical effective pressure due to higher cavity opening rates. We examine the contribution of mass transport, efficient drainage, and the linked-cavity negative feedback to surging. We also assess the likely bounds on poorly constrained subglacial hydrology parameters and adopt an ensemble approach to study their impact and interactions within those bounds.

We find that subglacial hydrology is an important system inductance for realistic ice stream surging but that the three formulations all exhibit similar surge behaviour within parametric uncertainties. Even a detail as fundamental as mass-conserving transport of subglacial water is not necessary for simulating a full range of surge frequency and amplitude. However, one difference is apparent: the combined positive and negative feedbacks of the linked-cavity system yield longer duration surges and a broader range of effective pressures than its poro-elastic and leaky-bucket counterparts.

## 1 Introduction

The role of subglacial hydrology at timescales longer than multiple decades and at ice sheet spatial scales is unclear. Previous studies have inferred that subglacial hydrology plays a strong role in internally (e.g. Siegfried et al., 2016) and externally (e.g. Joughin et al., 1996; Cook et al., 2021) driven ice sheet variability on sub-annual to multi-decadal timescales (Retzlaff and Bentley, 1993; Alley et al., 1994; Ou, 2021; Bennett, 2003). Observations beyond these timescales do not exist.

Several subglacial hydrologic systems have been conceptualized (Flowers, 2015). Constraint of the role of hydrological systems is further challenged by the large parametric uncertainties for all choices of drainage system. For example, the bounds of hydraulic conductivity vary over several orders of magnitude and according to the particular system (Werder et al., 2013). These uncertainties hinder widespread adoption of subglacial hydrology models in Earth systems models in general and glacial-cycle-scale ice sheet modelling in particular (Flowers, 2018). As such, what is needed to adequately

incorporate the subglacial hydrologic system into glacial cycle simulations is not understood.

We ask a basic question: does subglacial hydrology matter on longer than decadal timescales? And if so, to what extent are the structural details of the hydrological system important for this context, especially given the rest of the system uncertainties? Taking a modelling approach, we focus these broad questions on the following: is subglacial hydrology needed to capture Hudson Strait-scale ice stream cyclicity? If so, should effective pressure be dynamically determined – based on fully mass-conserving lateral drainage? Or does a zero-dimensional meltwater volume balance with a diagnostic pressure closure suffice? Turning to the parametric uncertainties, which are most important?

Previous model-based tests of Hudson Strait ice stream surging (e.g. Calov et al., 2010; Payne et al., 2000; Payne and Dongelmans, 1997; MacAyeal, 1993) have focused on thermomechanical feedbacks but omitted the contribution from the subglacial hydrological system. While these studies capture surges in their simulations based on these limited feedbacks, all models except one (model (d), Calov et al., 2010) implemented an abrupt transition at the frozen–temperate thermal boundary, suddenly initiating large-scale sliding within a grid cell. This abrupt thermal transition is physically unrealistic at the scales of ice sheet modelling: a region equivalent to a glacial-cycle-scale ice sheet model grid cell (100–2500 km$^2$) does not become instantly warm based with wholesale transition to basal sliding. Instead, the streaming portions of ice sheets transition to faster-sliding velocities as their coupling to the bed (effective pressure) decreases. Subglacial hydrology is a potentially critical piece of the binge–purge conceptual model of internal oscillations (MacAyeal, 1993) as heat production from sliding and deformation work generates meltwater.

Here we examine the contribution to ice sheet internal oscillations from the three most dominant forms of distributed subglacial hydrology – linked cavity (Schoof, 2010), poro-elastic (Flowers, 2000), and non-mass-transporting leaky bucket (Gandy et al., 2019) – relative to each other and to no hydrology at all. In each case, the frozen-to-temperate transition is smoothed following the work of Hank et al. (2023). We couple these processes to an ice sheet systems model, the Glacial Systems Model (GSM, Tarasov et al., 2012).

Simple configurations make system behaviours more interpretable (e.g. Calov et al., 2010; Payne et al., 2000). With a realistic bed and actual climate, spatio-temporal variations in model solutions are largely due to the variation in boundary conditions. We therefore model these coupled systems for a simplified North American analogue setup which implements a square bed and flat topography with soft beds in the southern latitudes and in the Hudson Strait and Bay area. The ice sheet is forced with a steady climate and first-order feedbacks: the northward-cooling temperature trend, vertical lapse rate, and thermodynamic moisture control. The numerical model retains important processes while still being feasi-

ble to run large ensembles over a glacial cycle on continental scales to probe parametric uncertainties.

Below, we first test the BAsal Hydrology Model (BrAHMs). This includes a demonstration of the mass conservation, convergence, and symmetry of BrAHMs2.0 and verification of its solutions against another prominent model, the Glacier Drainage System model (GlaDS, (Werder et al., 2013). Next we show the sensitivity of ice sheet geometry to subglacial hydrologic parameters in comparison with climate and ice sheet parameters. Finally, we compare results from a set of four large ensembles (between 11 000 and 20 000 members each) using no hydrology and linked-cavity, poro-elastic, and leaky-bucket hydrology.

## 2  Subglacial hydrology

In the context of continental-scale ice sheet modelling, resolving individual drainage elements and multiple topologies present within the domain is not computationally feasible. In this section we provide a brief overview of some structural choices made by others and present the options compared in this study, beginning with the current understanding of subglacial hydrology and progressing to increasingly approximate representations of it.

Water in the subglacial system flows either through inefficient drainage systems (pressure $\propto$ flux) or efficient drainage systems (pressure $\propto$ flux$^{-1}$, Flowers, 2015). Inefficient distributed networks are widespread under temperate areas of ice sheets, whereas efficient channel networks are discrete, localized elements. Each class evolves to the other and the change is controlled by system throughput, i.e. water flux. When the flux in an inefficient system rises above a threshold, the system transitions to efficient drainage. When the efficient system flux falls through different lower flux threshold, the system transitions to inefficient drainage, resulting in hysteresis (Schoof, 2010). Any mass-transporting hydrology model should have three main components: mass conservation describing transport, a flow law describing flux as a function of hydraulic gradient and subglacial water thickness, and a pressure closure relationship.

### 2.1  Inefficient flow

In the inefficient drainage regime, flux and water pressure rise together. Several inefficient drainage systems have been theorized: thin film, poro-elastic media, and linked cavities. Of these, poro-elastic and linked-cavity systems (e.g. Flowers, 2000; Walder, 1986) dominate recently published models (Flowers, 2015; de Fleurian et al., 2018), and As such, these are the two systems we model and contrast herein.

In the poro-elastic formulation, water can drain through the pore space of some permeable surficial material (e.g. till). Increasing subglacial water pressures expand the pore space and modify the permeability of the porous medium to flowing

water. The conceptual basis for this system is examined in greater detail by Flowers and Clarke (2002). The pressure closure has no theoretical basis and is based on a power law with empirically constrained parameters (Flowers, 2000).

In the linked-cavity system, cavities within the base of the ice open up as basal ice flows over and around bed protrusions – fast flow and larger objects beget larger cavities (Kamb, 1987). As cavities grow larger and numerous they form a connected network linked through smaller orifices giving a tortuous drainage network.

The substrate type that controls which inefficient system dominates – i.e. till cover and roughness – is variable (Pelletier et al., 2016; Brubaker et al., 2013). Conceivably, while poro-elastic drainage requires a porous ice sheet substrate, the cavities can form in any environment with bed protrusions which are less mobile than ice flow. A soft-bedded cavity has been seen at the base of a borehole in ice stream C (Carsey et al., 2002), and the theoretical basis for these cavities (Schoof, 2007) is motivated by drumlin formation (Fowler, 2009). However, cavities can only drain water once they grow enough to join and form a connected network (Rada and Schoof, 2018).

The contrast between kilometre-order or larger model scales and the metre-order or smaller process scales permits inefficient flow to be described as a continuum at the macro scale. On the macro scale, flux is related to water thickness and hydraulic gradient as follows (Flowers, 2015):

$$\boldsymbol{Q} = -k h_{\mathrm{wb}}{}^{\alpha} |\boldsymbol{\psi}|^{\beta-2} \boldsymbol{\psi}, \qquad (1)$$

with flux $\boldsymbol{Q}$, hydraulic conductivity $k$, and subglacial (basal) water thickness $h_{\mathrm{wb}}$. The gradient of the hydraulic potential is given by

$$\boldsymbol{\psi} = \nabla \left[ P_{\mathrm{water}} + \rho_{\mathrm{w}} g z_{\mathrm{b}} \right], \qquad (2)$$

with subglacial water pressure $P_{\mathrm{water}}$, density of freshwater $\rho_{\mathrm{w}}$, gravitational acceleration $g$, and basal topographic elevation $z_{\mathrm{b}}$. The exponents in Eq. (1) set laminar or turbulent flow. $\alpha = 1$ and $\beta = 2$ give Darcy's law for laminar flow through porous media (Darcy, 1856; Muskat, 1934). $\alpha = 5/4$ and $\beta = 3/2$ give the Darcy–Weisbach relation for turbulent flow through conduits (Clarke, 1996; Weisbach, 1855). Equations (2) and (1) are combined with a water pressure closure relationship given by the underlying physical system to get the formulations in Sect. 2.1.1 and 2.1.2.

Water sheet thickness is a continuum property used to describe the average amount of water in a grid cell. Changes in water thickness are given by the fluxes and the aggregate of sources and sinks, $m$, in the water transport Eq. (3):

$$\frac{\partial h_{\mathrm{wb}}}{\partial t} + \nabla \cdot \boldsymbol{Q} = m, \qquad (3)$$

where $\boldsymbol{Q}$ is the subglacial water flux, and $m$ is the aggregate of sources and sinks.

### 2.1.1 Poro-elastic system

Pressurized subglacial water flows through the pore space of a layer between ice and bedrock, conceptualized as the interstitial space between till grains. As water pressure increases, the permeability of the porous medium rises. Water pressure is related to subglacial water thickness by a non-linear function using pore-space saturation (Eq. 4). This poro-elastic drainage formulation is laid out in Flowers (2000). The flow law is Darcy's law describing laminar flux as a function of hydraulic gradient and subglacial water thickness. The pressure closure is an empirical relationship between the water column height in the elastic pore-space and subglacial water pressure.

The Darcy flow law is Eq. (1) with $\alpha = 1$ and $\beta = 2$. Water pressure in the elastic pore space is set by Eq. (4) (Flowers, 2000):

$$P_{\mathrm{water}} = P_{\mathrm{ice}} \left( \frac{h_{\mathrm{wb}}}{h_{\mathrm{c}}} \right)^{7/2}, \qquad (4)$$

where $P_{\mathrm{ice}}$ is the pressure due to the weight of overbearing ice and $h_c$ is the water thickness scalar interpreted as the thickness of the pore-space accommodating water.

### 2.1.2 Linked-cavity system

As ice flows over protrusions in the bed, cavities open in the lee side. The faster ice flows and the higher the protrusion, the greater the opening rate. The weight of the overbearing ice acts to close the void through viscous creep. The tradeoff between these two rates determines the net cavity size change rate. These cavities are linked through smaller connections and form a drainage network whose throughput is controlled by orifice size and system tortuosity. As water flows more quickly in the drainage network, wall melting due to frictional heating at the ice–water interface further opens cavities and the interconnecting orifices, forming a more efficient system. The Darcy–Weisbach flow law for turbulent flux depends on the hydraulic gradient and subglacial water thickness. The pressure closure is based on cavity opening and closing velocities and mass balance. The Darcy–Weisbach flow law is Eq. (1) with $\alpha = 5/4$ and $\beta = 3/2$ with $P_{\mathrm{water}} = P_{\mathrm{ice}} - N_{\mathrm{eff}}$. For the linked-cavity system, $k = k_{\mathrm{lc}}$ in Eq. (1) aggregates quantities such as tortuosity, hydraulic gradient across the orifice, and cavity density. Completing the set of equations, the effective pressure, $N_{\mathrm{eff}}$, is given by the opening–closing relationship for the cavity cross-sectional area with respect to time in Eq. (5). This has three parts:

– wall melting term ($\propto \boldsymbol{Q} \cdot \boldsymbol{\psi}|$),

– opening from sliding over bed protrusions ($\propto u_{\mathrm{b}} h_{\mathrm{r}}$),

– closing due to overburden pressure (creep) ($\propto N_{\mathrm{eff}}^{n} S$).

$$\frac{\partial S}{\partial t} = c_1 \boldsymbol{Q} \cdot \boldsymbol{\psi} + u_b h_r - c_2 N_{\text{eff}}^n S, \tag{5}$$

where $S$ is the cavity size, $c_1$ and $c_2$ are constants, $\boldsymbol{Q}$ is flux, $u_b$ is basal sliding velocity, $h_r$ is bed protrusion height, and $n$ is the exponent from Glen's flow law (Schoof, 2010; Werder et al., 2013). These three terms act to increase or decrease cavity area.

## 2.2 Efficient flow

In the efficient drainage regime, flux and water pressure are inversely related. Flux in the efficient system occurs in subglacial tunnels incised into overbearing ice (Röthlisberger, 1972), down into subglacial sediments (Walder and Fowler, 1994), or into hard bedrock (Alley, 1989). Channels eroded into bedrock remain in the same place through time, while those formed in ice or sediment can open, move, and close depending on overbearing ice and hydrologic conditions. The most commonly modelled efficient system is the Röthlisberger channel (R channel) carved up into the overbearing ice (de Fleurian et al., 2018). Dendritic subglacial tunnels open up into the ice from the base by wall melting due to frictional heat from the contact between ice and flowing water (Röthlisberger, 1972) – the faster the water, the larger the channel. Counter to the inefficient regime, water pressure and flux are inversely proportional (Schoof, 2010; Flowers, 2015). As water percolating through the inefficient system flows quickly enough to give significant wall melting, the system becomes unstable and quickly transitions to a channelized system (Schoof, 2010). Schoof (2010) showed that Eq. (5) bifurcates into the inefficient linked-cavity system and the efficient R-channel system, the switch between the two controlled by flux in the subglacial system. At high fluxes, frictional melting of the tunnel ice wall from fast-flowing water becomes a runaway effect opening an R channel into the ice. Canals likely open due to high flux as well in the subglacial system, where energetic water mobilizes sediment along its path (Alley, 1992; Walder and Fowler, 1994).

The conceptual basis for the efficient flow model herein is the R channel, which evolves out of the inefficient system based on high fluxes.

## 3 Model description

The model used here is a fully coupled system of hybrid SIA/SSA (shallow ice approximation, shallow shelf approximation) ice physics (Pollard and DeConto, 2012) and 3D ice thermodynamics and 1D bed thermodynamics (Tarasov and Peltier, 2007). The climate forcing imposes a background surface temperature trend and elevation dependencies for temperature and precipitation. The subglacial hydrology model includes a choice of linked-cavity, poro-elastic, or leaky-bucket inefficient drainage, of which the linked-cavity and poro-elastic drainage can be coupled to the efficient drainage tunnel solver. The transition from frozen to temperate is smoothed to more realistically capture the transition to sliding (as in model (d) of Calov et al., 2010) following the work of Hank et al. (2023). A more detailed description of the GSM is forthcoming (Tarasov et al., 2023).

## 3.1 Subglacial hydrology model

The subglacial hydrology model – BrAHMs2.0 – is an extensive update to version 1.0 (Kavanagh and Tarasov, 2018). The update includes the addition of linked-cavity and leaky-bucket systems, an updated generalized grid, modified convergence criteria, a modified flux limiter, and code restructuring. This model uses a finite-volume discretization with a staggered Arakawa C grid (fluxes at interfaces; Arakawa and Lamb, 1977). In the case of the 2D mass-transporting hydrology setups (poro-elastic and linked cavity), we implement the generalized flux calculation in Eq. (1) with a choice of either the pressure-determining closure of Flowers (2000) or a modified version of Schoof (2010) as in that of Werder et al. (2013) and Bueler and van Pelt (2015), whereby the cavity opening rate is proportional to the difference in bed roughness and subglacial water sheet thickness. Schoof (2010) shows that the wall melting term in Eq. (5) is unimportant until a critical value is reached and the runaway effect opens tunnels (see assumption 1 below). As such, the wall melting term is assumed to be 0 until tunnelling is triggered.

$$\frac{\partial S}{\partial t} = u_b h_r - c_2 N_{\text{eff}}^n S. \tag{6}$$

In this model the cavities are described as a continuum: height of a cavity averaged over protrusion spacing ($l_r$) is given as $h_{\text{cav}} = \frac{S}{l_r}$:

$$\frac{\partial (h_{\text{cav}} \cdot l_r)}{\partial t} = u_b h_r - c_2 N_{\text{eff}}^n h_{\text{cav}} \cdot l_r. \tag{7}$$

The opening term is modified to drop as average cavity thickness rises over the bed protrusion $u_b (h_r - h_{\text{cav}})$ as in (e.g.) Werder et al. (2013), and cavities are assumed filled by subglacial water ($h_{\text{cav}} = h_{\text{wb}}$; see assumption 3). This leads to the relationship for water pressure evolution:

$$\frac{\partial P_w}{\partial t} = \frac{\rho_w g}{\phi_{\text{eng}}}$$
$$\left( -\nabla \cdot \boldsymbol{Q} + m_t - u_b (h_r - h_{\text{wb}}) / l_r + c_2 [P_{\text{ice}} - P_w]^n \right), \tag{8}$$

where $\phi_{\text{eng}}$ is the englacial porosity and $m_t$ Eq. (8) is derived in Appendix E1 following Bueler (2014) and is similar to that used in Werder et al. (2013), Hewitt (2013), and Bueler and van Pelt (2015).

While in the linked-cavity model the hydraulic conductivity is a single parameter, in the poro-elastic model Flowers (2000) uses a meltwater-thickness-dependent arctan function for hydraulic conductivity to capture a transition from low to

**Table 1.** Table of parameter names, descriptions, their numerical ranges, and the subglacial hydrologic system they parameterize used in the ensembles for this study. LC corresponds to the linked-cavity system, PE to the poro-elastic system, and LB to the leaky-bucket system.

| Name | Description | Range | Drainage |
|---|---|---|---|
| $k_{\text{lc}}$ | Hydraulic conductivity of cavities | $1.00 \times 10^{-6}$–$1.00 \times 10^{1}$ | LC |
| $h_{\text{r}}$ | Vertical basal roughness height | $1.00 \times 10^{-2}$–$2.00 \times 10^{1}$ | LC, tunnel |
| $R_{\text{ratio}}$ | Roughness height : wavelength ($h_{\text{r}}/l_{\text{r}}$) | 1.0–20.0 | LC |
| $Q_{\text{scale}}$ | Tunnel switch criterion scaler | $1.00 \times 10^{-3}$–$1.00 \times 10^{0}$ | Tunnel |
| $F_{N_{\text{eff}}}$ | $N_{\text{eff}}$ normalization in sliding | $1.0 \times 10^{4}$–$1.0 \times 10^{6}$ | LC, PE, LB |
| $T_{\text{froz}}$ | Freeze point, hydrology system | $-1.00 \times 10^{-0}$–$0.00 \times 10^{0}$ | LC, PE, LB |
| $k_{\text{max}}$ | Max hydraulic conductivity | $1.00 \times 10^{-6}$–$1.00 \times 10^{1}$ | PE |
| $k_{\text{ratio}}$ | Max : min hydraulic conductivity | $1.00 \times 10^{0}$–$1.00 \times 10^{2}$ | PE |
| $h_{\text{c}}$ | $h_{\text{wb}}$ quotient, water pressure | $1.00 \times 10^{-1}$–$5.00 \times 10^{1}$ | PE, LB |
| $s_{\text{drain}}$ | Drainage rate | $1.00 \times 10^{-3}$–$1.00 \times 10^{-2}$ | LB |

**Table 2.** Table of subglacial hydrology configurations showing the drainage law used, whether the efficient drainage system is coupled in, and what effective pressure is used. LC corresponds to the linked-cavity system, PE to the poro-elastic system, and LB to the leaky-bucket system. TS1

| Configuration | Drainage | Efficient drainage | $N_{\text{eff}}$ |
|---|---|---|---|
| LC | Darcy–Weisbach | yes | Dynamic, $u_{\text{b}}$ two-way feedback; Bueler and van Pelt (e.g. 2015) |
| PE | Darcy | yes | Diagnostic (e.g. Flowers, 2000) |
| LB | None | no | Diagnostic (e.g. Flowers, 2000) |
| NH | n/a | n/a | n/a |

n/a: not applicable

high permeability during the expansion of the pore space:

$$\log(k) = \frac{1}{\pi}\left[\log(k_{\text{max}}) - \log(k_{\text{min}})\right]$$
$$\arctan\left[k_a\left(\frac{h_{\text{wb}}}{h_{\text{c}}} - k_b\right)\right] + \frac{1}{2}\left[\log(k_{\text{max}}) + \log(k_{\text{min}})\right], \quad (9)$$

where $k$ is the poro-elastic hydraulic conductivity, $k_{\text{max}}$ is the maximum conductivity, $k_{\text{min}} = k_{\text{max}}/k_{\text{ratio}}$ is the minimum conductivity, $h_{\text{c}}$ is the critical water thickness in the pore space ($h_{\text{c}}$ in Table 1), and $k_a$ and $k_b$ control the transition between the maximum and minimum conductivity.

Numerically, hydraulic conductivity in both the poro-elastic and linked-cavity formulations is defined at the cell centres and is a function of cell temperature relative to the pressure melt point ($T_{\text{bp}}$). To account for the transition from fully cold based (frozen) to fully warm based (thawed), the bed is assumed to be fully frozen below $T_{\text{froz}}$ and the hydraulic conductivity is given the following dependence on basal temperature relative to the pressure melt point ($T_{\text{bp}}$):

$$k_{\text{therm}}^{i,j} = (k - k_{\text{f}})$$
$$\cdot \left(1 - \exp\left[3[\![\min\left(0.0, T_{\text{bp}}^{i,j}\right), T_{\text{froz}}]\!]/T_{\text{froz}} - 1\right]\right) + k_{\text{f}}, \quad (10)$$

where $k_{\text{f}}$ is the hydraulic conductivity of frozen till (effectively zero). As the flux should be a function of the potential

difference across the interface, the harmonic mean of the adjacent cell-centred conductivities gives the most appropriate interface conductivity (Patankar, 1980).

$$k_{\text{we}}^{ij} = \frac{2k_{\text{therm}}^{i-1j}k_{\text{therm}}^{i,j}}{k_{\text{therm}}^{i-1j} + k_{\text{therm}}^{ij}} \quad (11)$$

In order to assess the importance of transport vs. pressure determination in surging, we implement a non-mass-conserving zeroth-order leaky-bucket scheme: a constant drainage rate ($s_{\text{drain}}$) counters the melt rate ($s_{\text{melt}}$) to give basal water thickness in that cell, following Gandy et al. (Eq. 3, 2019):

$$\frac{\partial h_{\text{wb}}}{\partial t} = s_{\text{melt}} - s_{\text{drain}}. \quad (12)$$

The leaky-bucket scheme uses the empirical pressure-determining closure of Flowers (2000) shown in Eq. (4), with basal water thickness limited between 0 and the critical thickness of the pressure closure ($h_{\text{c}}$ in Eq. 4).

Fully modelling the process of efficient drainage of water through the channel system would require very short time steps due to Courant–Friedrichs–Lewy (CFL) (Courant et al., 1928) restrictions and consequently prohibitively long run times. Given the disparity in timescale between efficient drainage (sub-annual) and the dynamical behaviour examined here (centennial- to millennial-scale surging), it is unlikely that dynamically versus diagnostically modelling the

efficient drainage will have an effect on the longer timescale surging, though as the model is non-linear there is potential for propagation across timescales. As such, an alternate scheme is used under the assumption that drainage happens far quicker than in the inefficient system (assumption 2), which should especially hold for ice sheet modelling contexts. If flux at a cell face exceeds the bifurcation threshold or "critical discharge" of Schoof (2010), water is routed down the background hydraulic gradient (i.e. from topography and ice sheet overburden), filling in potential lows along the way until routed water is depleted or exits the ice sheet. This subglacial meltwater routing scheme is a slight modification of the downslope surface meltwater routing scheme of Tarasov and Peltier (2006) (i.e. with a modified hydraulic gradient). This routing scheme is further discussed in Kavanagh and Tarasov (2018).

For details on the numerical solver used here, readers are invited to read Appendix B. Assumptions used in the design of this model are examined in Appendix C. The verification of the model implementation presented in Appendix D shows that the model

1. gives symmetric solutions given symmetric boundary conditions,

2. converges under increasing spatial and temporal resolution at a rate commensurate with the discretization schemes,

3. conserves mass,

4. gives similar solutions to another model using similar physics (GlaDS, Werder et al., 2013).

### 3.2 Basal drag coupling

The basal velocity is from either a hard- or soft-bed sliding rule. For the hard bed the basal sliding rule is a fourth-power Weertman sliding law (Cuffey and Paterson, 2010):

$$u_b^{hard} = \frac{c_{hard} c_{fslid} F_{N_{eff}} |\tau_b|^3 \tau_b}{N_{eff}}, \tag{13}$$

where $c_{hard}$ is a parameterized sliding coefficient which includes a parameterization for basal roughness, $F_{N_{eff}}$ is the effective pressure normalization factor, $N_{eff}$ is the effective pressure given by the subglacial hydrology model, and $\tau_b$ is the basal drag. The basal velocity for soft-bedded sliding is similarly a Weertman-type sliding law with integer exponent values between 1 and 7.

$$u_b^{soft} = \frac{c_{soft} c_{rmu} F_{N_{eff}} |\tau_b|^{b_{till}-1} \tau_b}{N_{eff}}, \tag{14}$$

with separately parameterized soft-sliding coefficient $c_{soft}$ (which also includes a parameterization for basal roughness).

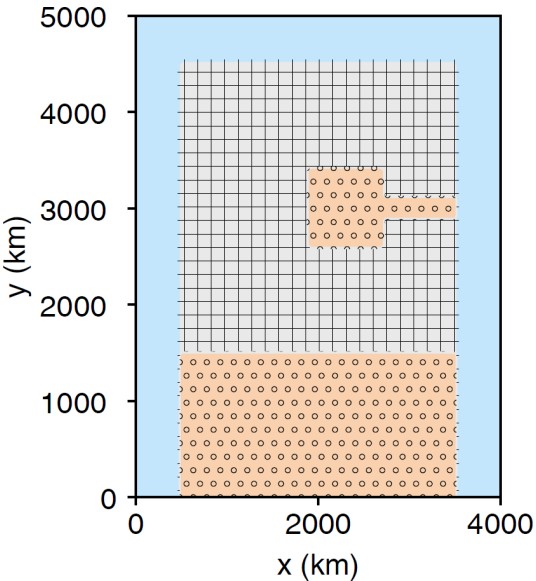

**Figure 1.** This map of the LISsq bed configuration shows the extent of the domain and the portion of the Hudson Bay and Strait and southern soft beds. Gray hatched regions are hard bedded, beige dotted regions are soft bedded, and blue represents water where ice is ablated.

## 4   LISsq experimental design

Using a simple setup without externally driven variability from topography, a complex land–sea mask, and an unsteady climate, system behaviour is due to the initial transient response and internal feedbacks. Our Laurentide Ice Sheet square (LISsq) setup includes broad features of the North American bed (Fig. 1) and computationally cheap first-order diagnostic climate imposed as a steady forcing with ice sheet thickness feedbacks. The simple climate allows a free southern margin determined by the background temperature and feedbacks, giving a dynamically determined ice sheet geometry at 50 km horizontal resolution. Next we present the design choices of this setup in three categories: bed, climate, and glacial systems.

### 4.1   Bed

LISsq aims to probe the effect of large-scale hard- to soft-bed transitions characteristic of North America. This simplified setup allows separating out the internal feedbacks from the externally forced elements (e.g. variability from real topography and a land–sea mask and unsteady, spatially varying climate). The shorter run times of this setup also allow larger ensembles, giving a better probe of the parameter space. The simplicity helps with model verification as any variability in the model stems purely from the encoded physical processes.

The majority of the inferred Late Pleistocene Laurentide substrate has been hard bedded (Clark et al., 2006), with unconsolidated sediment cover at the south and in the Hud-

son Bay and Strait. The Heinrich Event Intercomparison (HEINO) experiments were conducted over similar length-scale hard beds with the same soft-bedded Hudson Bay and Strait at the centre of the hard bed (Calov et al., 2010). HEINO differed in that it included a circular continental configuration bounded by a highly ablating ocean – the ice sheet geometry was largely set. Here we wish to examine surge behaviour for a variety of ice sheet geometries within the roughly approximate range of the Laurentide length scales and bed. As such, a rectangular bed geometry is set with the boundary of the soft-bedded south at a constant latitude and an equilibrium line which is free to evolve with a changing ice sheet.

## 4.2 Climate

The LISsq climate prescribes a linear background temperature trend with lapse rate feedback. The annually averaged surface temperature, $T_{surf}$, is

$$T_{surf} = T_{north} + [\![0, T_{grad}(5000 - y)]\!] - LH, \tag{15}$$

where $T_{north}$ is the ground level temperature at the northern end (°C), $T_{grad}$ is the latitudinal warming rate (°C km$^{-1}$), $L$ is the slope temperature lapse rate (°C km$^{-1}$), and $H$ is ice sheet thickness (m; recall that the bed is at constant elevation and glacial isostatic adjustment is not included). The brackets, $[\![\ ]\!]$, denote the maximum.

These temperatures are then used together with a positive degree day scheme (PDD) to simulate net seasonal contribution to accumulation and ablation for an annual average temperature. The positive degree day sum assumes a 100 d melt season length with temperatures 10 °C warmer than the annual mean, $T_{surf}$, and a melt coefficient in m per PDD. Ablation is then

$$\dot{b}_{melt} = C_{pdd} [\![0.0, 100 (T_{surf} + 10)]\!], \tag{16}$$

where $b_{melt}$ is ablation (m yr$^{-1}$) and $T_{surf}$ is surface temperature (°C). Accumulation incorporates the thermodynamic effect on atmospheric moisture content using the August–Roche–Magnus approximation for the Clausius–Clapeyron relationship (Lawrence, 2005) with parameter ranges adjusted for undersaturated air. Accumulation, $b_{accum}$, is 0 where $T_{surf} \geq 0$ °C:

$$\dot{b}_{accum} = p_{ref} \, e^{h_{pre} T_{surf}}, \tag{17}$$

where the reference precipitation rate, $p_{ref}$, and precipitation pre-exponential factor, $h_{pre}$, are ensemble parameters.

## 4.3 Glacial systems

We use a subset of the full-featured GSM for this setup. Here we omit glacial isostatic adjustment, surface meltwater drainage, sediment transport and production, and ice shelves with grounding-line flux and calving model. This is in order to clearly show the effect of hydrology feedbacks on ice flow and ice thermomechanics.

## 4.4 Parameter range estimation

In this section we justify chosen parameter ranges based on physical and heuristic arguments and current understanding in the literature.

## 4.5 Hydraulic conductivity parameterization

The range of values appropriate for hydraulic conductivity varies according to whether the drainage system is assumed to be poro-elastic or linked cavity or whether the flux is assumed to be laminar (Darcian) or turbulent (Darcy–Weisbach). Hydraulic conductivity is not truly known at the continuum-level macro-scale. Here we use a range based on bounding subglacial hydrologic flow velocities, typical hydraulic gradients, and subglacial water thicknesses.

The velocity of water flow in the subglacial channel end-member imposes an upper bound on the linked-cavity end-member flow velocity in the bifurcated channel–linked-cavity system (Schoof, 2010). Chandler et al. (2013) used dye tracing experiments at a land-terminating west Greenland catchment to measure maximum velocities between moulin injection site and the margin. Their slowest first arrival time gave $1.00 \, \mathrm{m\,s^{-1}}$ in the efficient drainage regime.

Fast ice velocities (e.g. $\approx 1 \, \mathrm{km\,yr^{-1}}$) give a loose lower bound on water flow speeds. Whereas the viscosities differ by many orders of magnitude ($10^{14}$ Pa s for ice versus $10^{-3}$ Pa s for water; Cuffey and Paterson, 2010), the pressure gradient forces are less dissimilar. Assuming the hydraulic gradient is approximately equivalent to that imposed by ice sheet and bed topography (i.e. no contribution from basal water pressure) – around 1000 m per 56 km (Chandler et al., 2013) ice sheet surface gradient contribution and 500 m per 56 km for bed contribution (Morlighem et al., 2013) – gives a hydraulic gradient of $\psi \approx 240 \, \mathrm{Pa\,m^{-1}}$. Assuming further ranges of 1 mm to 10 m of basal water thickness and Darcy–Weisbach flow speeds between $3 \times 10^{-4}$ and $1 \times 10^0$ m s$^{-1}$ gives a range of linked-cavity hydraulic conductivity ($k_{cond}$, Table 1) between $1 \times 10^{-5}$ and $1 \times 10^{-1}$ m$^{5/4}$ Pa$^{-1/2}$ s. To ensure complete bounding, we probe a wider range of $1 \times 10^{-6}$ and $1 \times 10^1$ m$^{5/4}$ Pa$^{-1/2}$ s. This range encapsulates values suggested by Hager et al. (2022) and Werder et al. (2013). Flowers (2000) assessed the range of hydraulic conductivities to be $k_{max} = 1 \, \mathrm{m\,s^{-1}}$ and $k_{min} = 10 \times 10^{-7} \, \mathrm{m\,s^{-1}}$. The hydraulic conductivity transitions from $k_{max}$ to $k_{min}$ according to Eq. (9).

## 4.6 Basal roughness

The height of bedrock protrusions relevant to subglacial cavity formation and its spatial variation lacks assessment in the

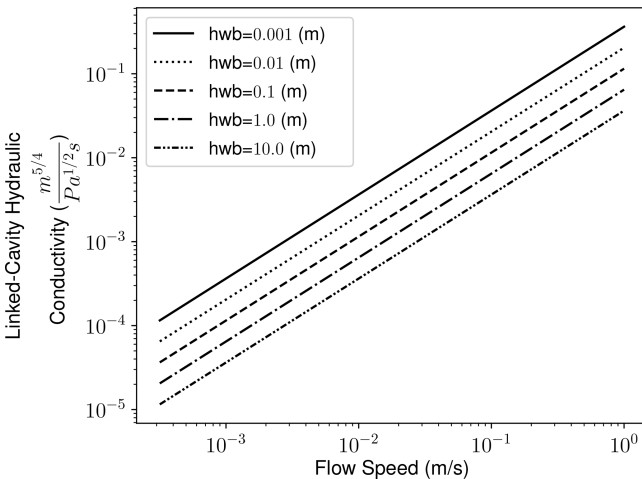

**Figure 2.** Range of linked-cavity hydraulic conductivities based on basal water flow speed, hydraulic gradient, and basal water thickness ranges in the text – using $k = \frac{v}{\psi^{1/2}h^{1/4}}$.

literature and justified values are difficult to come by. The height of these protrusions, or terrain roughness, affects several basal processes in glaciated regions, including heat generation in basal ice, sliding, subglacial cavity opening, and bedrock quarrying. Length scales relevant to subglacial cavity formation have been estimated from chemical alteration of bedrock (the deposition of calcium carbonate precipitates) (Walder and Hallet, 1979). These cavity outlines form during sliding-associated regelation when water refreezes at the glacier substrate on the lee side of bedrock highs, precipitating dissolved carbonates. The deposits in this study indicate cavities 0.1–0.15 m high. Several studies then use a value in this range (e.g. Werder et al., 2013). Kingslake and Ng (2013) refers to Walder (1986) for this value, but Walder (1986) does not provide any justification for it in their Table 2 and does not refer explicitly to the earlier work of Walder and Hallet (1979).

In deglaciated areas with bed access, quantifying roughness at the ice sheet scale is a non-unique problem and measures abound. For example: standard deviation of elevation, power spectral density of elevation, and local bed slope. These are relative measures which do not identify the typical prominence of roughness features in a domain. What is needed for modelling linked cavities is the average height of bedrock protrusions relevant to the cavity scale (itself uncertain) at given wavelengths. How these heights vary spatially for previously glaciated regions has not been assessed. Identifying this as a gap in the current glaciological literature, we adopt similar scale values and probe a wide range in order to capture ice sheet sensitivity to the scale of cavity-forming bump height. As stated above, Werder et al. (2013) and Kingslake and Ng (2013) both use $h_r = 0.1$ with the latter referring to Walder (1986), who gives a range of 0.01 to 0.5 m for the relevant bump height. Iverson (2012) show

cavities and quarrying are intrinsically linked. As such, the step size of quarried surfaces may indicate a scale for cavity growth. Anderson et al. (1982) mapped cavities forming along 1 m high steps at the base of Grinnell Glacier in Montana, United States. Following the same reasoning, the size of quarried boulders also gives an estimate of the upper bound for length scales. Though less common, 20 m boulders can be found (though if transported debris were comminuted in transit, the original size distribution would have been larger). As such, we use a range of $h_r \in [0.01, 20.0]$ m and a range for the roughness wavelength as a function of roughness of $l_r \in [1.0, 20.0] \times h_r$.

### 4.7 Hydrology temperate transition

This parameter is used to interpolate between a conducting (at $0\,°C$) and non-conducting (at a lower-bound temperature) hydrologic system with a logic similar to the temperature ramp reasoning. Thus, the range is based on Hank et al. (2023) and the lower bound of the interpolation is probed in the range of $[-1.0, 0.0]$.

### 4.8 Tunnel-switching scalar

The flux threshold switch from inefficient to efficient drainage is given by the ratio of cavity opening due to sliding versus wall melting from viscous heating (Schoof, 2010):

$$Q_{crit} = Q_{scale} \frac{u_b h_r / l_r}{c_1(\alpha - 1)\psi}, \tag{18}$$

where $u_b$ is the velocity, $h_r / l_r$ the basal roughness ratio, $c_1$ a scalar, $\alpha$ the Darcy–Weisbach water thickness exponent, $\psi$ the hydraulic gradient. $Q_{scale}$ is a scale factor adjusting for subgrid uncertainty – small-scale fluctuations in flux may trigger a runaway tunnelling positive feedback affecting the larger scale.

### 4.9 Effective pressure normalization

This is the value used to normalize the effective pressure in the basal sliding velocity calculation and is set based on typical effective pressures. Effective pressures greater than this parameter values should slow sliding and less than that should hasten sliding. We set this range to $[10\,kPa, 1\,MPa]$ based on the typical effective pressure values seen in Fig. 10. The effective pressure and normalization ($F_{N_{eff}}$) is incorporated into the hard and soft basal sliding velocities in Eqs. (13) and (14).

### 4.10 Basal sliding parameters

The soft and hard sliding factors used in Eqs. (13) and (14) were set to wide bounds somewhat outside the recommended range for the GSM (Tarasov et al., 2023); the power for soft-bedded sliding was kept within the typical range. These ranges were $c_{rmu} \in [0.01, 4.0]$ (set such that sliding speed

is $\approx 3\,\mathrm{km\,yr^{-1}}$ for $(30\,\mathrm{kPa})^{b_{\mathrm{till}}}\,\mathrm{kPa}$ of basal drag), $c_{\mathrm{fslid}} \in [0.0, 5.0]$ (set such that sliding speed is $\approx 200\,\mathrm{m\,yr^{-1}}$ for $100\,\mathrm{kPa}$ of basal drag), and $b_{\mathrm{till}} \in [1, 7]$.

## 4.11 Climate parameters

A range of $[5, 10]\,^{\circ}\mathrm{C\,km^{-1}}$ is used for slope lapse rate on the basis of PMIP2 Greenland model simulations in Erokhina et al. (2017). The range for $T_{\mathrm{north}}$ was obtained from the PMIP4 ensemble mean distribution of northern ($> 75^{\circ}$) latitude temperatures at the Last Glacial Maximum (LGM) in Kageyama et al. (2021) shown in Fig. 3. The precipitation parameter ranges in Eq. (17) were adjusted to bound the range of precipitation and temperatures below freezing in Kageyama et al. (2021), as shown in Fig. 3.

## 4.12 Ensemble design and parameter sensitivity

To understand the effect of hydrology, ensembles for different model configurations are compared: linked-cavity (LC), poro-elastic (PE), leaky-bucket (LB), and no-hydrology (NH) ensembles – 18 816, 19 992, 15 288, and 11 760 runs in each ensemble respectively. Each ensemble varied the hydrology, ice sheet, and climate parameters simultaneously in order to capture parameter interactions and the number of runs was scaled with the number of parameters in each setup (15 in LC, 16 in PE, 12 in LB, and 9 in NH, shown in Table 1). The parameter space is sampled with the quasi-random low-discrepancy Saltelli extension of the Sobol sequence (Saltelli, 2002) as implemented in SALib (Herman and Usher, 2017) with second-order terms enabled. Parameters are sampled with a log-uniform distribution for parameter values which vary over orders of magnitude. Each run proceeded for $100\,\mathrm{kyr}$ with the first $50\,\mathrm{kyr}$ taken as spin up (from no ice, initial accumulation given by the background temperature from $T_{\mathrm{north}}$ and $T_{\mathrm{grad}}$). Ensembles were run on a heterogenous Linux cluster with 24–32 Gb RAM and 8–24 Xeon or Opteron cores per node, clock speeds ranging from 2.4 to 2.7 GHz and a total of 652 cores. Run times averaged about 3 h.

Ice sheet geometries vary widely among runs for all model configurations. Maximum ice thickness ranges from 0 to $\sim 6000\,\mathrm{m}$, while the maximum north–south extent ranges from 0 to $4500\,\mathrm{km}$. Here we study surge behaviour at a scale similar to the Laurentide ice sheet by sieving (discarding runs outside the target metric ranges) the ensembles according to maximum ice sheet thickness and north–south extent. At the Last Glacial Maximum (LGM, 22 ka), the maximum ice thickness was plausibly around $4000\,\mathrm{m}$ (Tarasov et al., 2012). We use this estimate with a lower bound of $3000\,\mathrm{m}$ for the sieve in the main study and examined additional sieves with bounds $[2500, 3500]\,\mathrm{m}$ and $[3500, 4500]\,\mathrm{m}$ in CE1 Appendix A. LGM north–south extent was $\approx 4000\,\mathrm{km}$, while the last margin to fully encircle the Hudson Bay and Strait

(11.50 ka) extended $\approx 2500\,\mathrm{km}$ north to south (Dalton et al., 2020).

The importance of hydrology parameters for determining ice sheet geometry can be probed with sensitivity analysis. Local sensitivity analysis methods neglect interaction terms important for studying feedbacks in coupled models and so are not applicable here (Saltelli et al., 2008). Meanwhile variance-based (Sobol, 2001) methods require assumptions about the sampling structure of the underlying inputs. The trouble with coupled models is that they can be unstable; as such, there are incomplete runs which render sampling structure assumptions moot. There are other non-parametric sensitivity methods which do not require assumptions about the input sample distribution, but these require sample sizes even larger than those presented here in order to converge (e.g. Borgonovo, 2007; Pianosi and Wagener, 2015).

We develop a novel non-parametric method to measure sensitivity: we assess ice sheet geometry sensitivity to parameters by comparing the original uniform input parameter distribution with the parameter distribution corresponding to the sieved geometries (limiting the ensemble to those within geometric bounds). The non-parametric nature alleviates the need to make assumptions about the underlying parametric distribution class (e.g. variance is a normal distribution parameter). Using the impact of a sieve on parameter distribution to measure parameter sensitivity means that assumptions about the sampling methodology are not required and that successive sieves can be applied to the ensembles to measure different aspects of model sensitivities. For example, in Sect. 5.1 we measure the sensitivity of surge frequency for those ensemble members which pass the geometry sieve by further sieving on surge frequency. Parameters which do not control the ice sheet geometry will have a similar distribution after selecting for that geometry range as the original input sample distribution. The more modified the distribution, the more sensitive the parameter. More precisely, each distribution is approximated with a kernel density function (KDF) normalized to unit area under the KDF. The sensitivity metric is then the integral of the absolute difference between the sieved and unsieved KDFs, i.e. the measure of how much the sieve modifies each parameter's KDF. For example, the maximum KDF difference would stem from a narrow spike on the sieved distribution, which would mean that parameter strongly controls the model output, e.g. the more limited range indicated for $h_{\mathrm{pre}}$ in Fig. 4a. We add a uniformly sampled dummy parameter not used by the model to set a threshold of accuracy of the sensitivity metric in each case. This dummy parameter has a very similar input and sieved distribution (with minor differences due to the essential random sampling from sieving), for example that for the LC geometry sieving in Fig. 4.

The sensitivity metrics for all parameters in Fig. 5 rise above the baseline significance level set by the dummy parameter in each ensemble. The temperature coefficient in the August–Roche–Magnus relation ($h_{\mathrm{pre}}$), north–south temperature gradient and intercept ($T_{\mathrm{grad}}$ and $T_{\mathrm{north}}$, Eq. 15), and

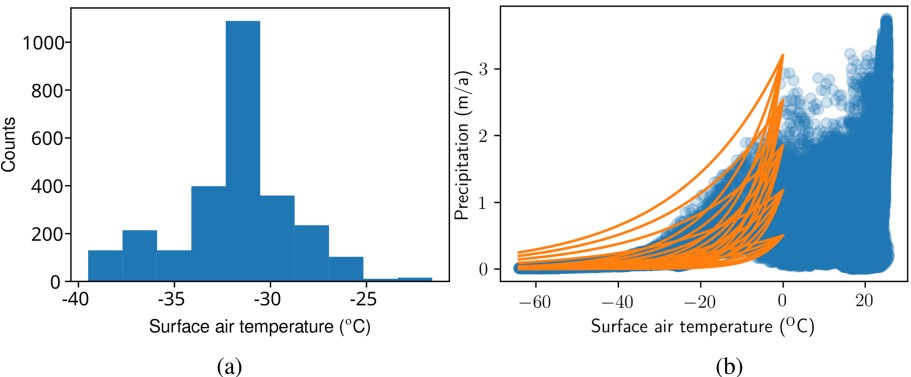

(a)                                                                      (b)

**Figure 3.** Precipitation and temperature values extracted from PMIP4 (Kageyama et al., 2021) ensemble mean fields at LGM. A histogram of surface air temperatures (count of points north of 75° N with temperature in the given bin) is shown in **(a)**. A scatter plot of precipitation and surface air temperature with overlaid precipitation temperature relationships showing the range of parameterizations used is presented in **(b)**.

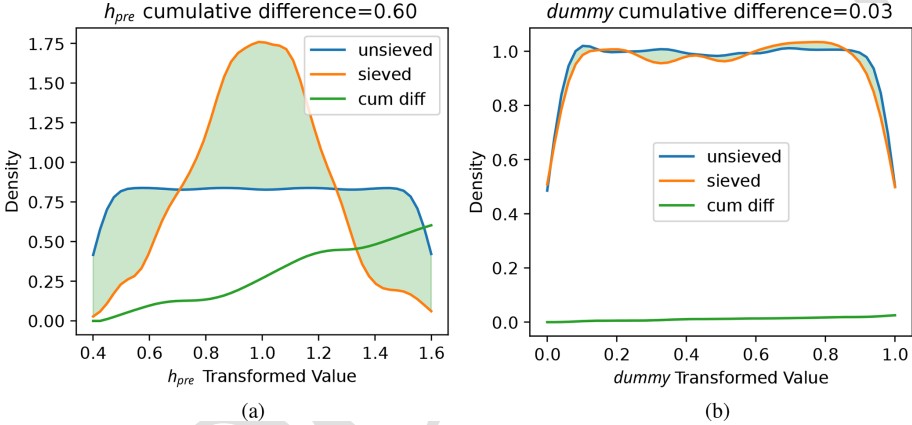

(a)                                                                      (b)

**Figure 4.** Cumulative kernel density function difference sensitivity metric for the most sensitive parameter, $h_{\mathrm{pre}}$ **(a)**, and the least sensitive parameter, dummy **(b)**, for the LC setup ice sheet geometry sieve. The parameter values are transformed for input to the GSM. The blue line shows the ensemble total parameter value distribution, orange shows the distribution after sieving the ensemble for geometry, and the green line shows the cumulative (integrated) absolute difference between blue and orange up to that value. The total cumulative difference gives the sensitivity measure, shaded in green.

lapse rate are the top four geometry-controlling parameters in all cases except LB (though lapse rate is close for this ensemble as well). In the hydrology enabled setups, hydrology parameters rank in the top five.

The ranked parameter sensitivity for each model in Fig. 5 exhibits an inflection point in parametric sensitivity which we use to determine the number of controlling parameters. This inflection point is an approximate indication of the diminishing sensitivities in the model setup. As such, parameters to the right of this point are taken as sensitive and those to the left are considered insensitive and could be fixed for the purposes of geometry. Around two-thirds of parameters fall on the right-hand side of the inflection in each ensemble. For those hydrology-bearing model configurations, half or more of the hydrology parameters lie in the sensitive zone.

This shows that subglacial hydrology is even important at the scale of whole ice sheet geometry.

The most influential hydrology parameter in the LC setup is hydraulic conductivity, which controls the dynamic effective pressure, while in LB and PE the geometry is quite sensitive to the normalization of the effective pressure in basal drag (though PE is more sensitive to the tunnelling tendency, $Q_{\mathrm{scale}}$). In the LC case, the ice sheet geometry is most sensitive to those parameters which control the dynamics of effective pressure themselves ($k_{\mathrm{cond}}$ and $h_{\mathrm{r}}$). In the PE case, the parameters controlling the transition to efficient drainage ($Q_{\mathrm{scale}}$) and effective pressure normalization are most important hydrology parameters. These parameters are both diagnostic controls on subglacial water balance and sliding velocity. Similar to PE, the most important subglacial hydrology parameter for LB is the effective pressure normalization.

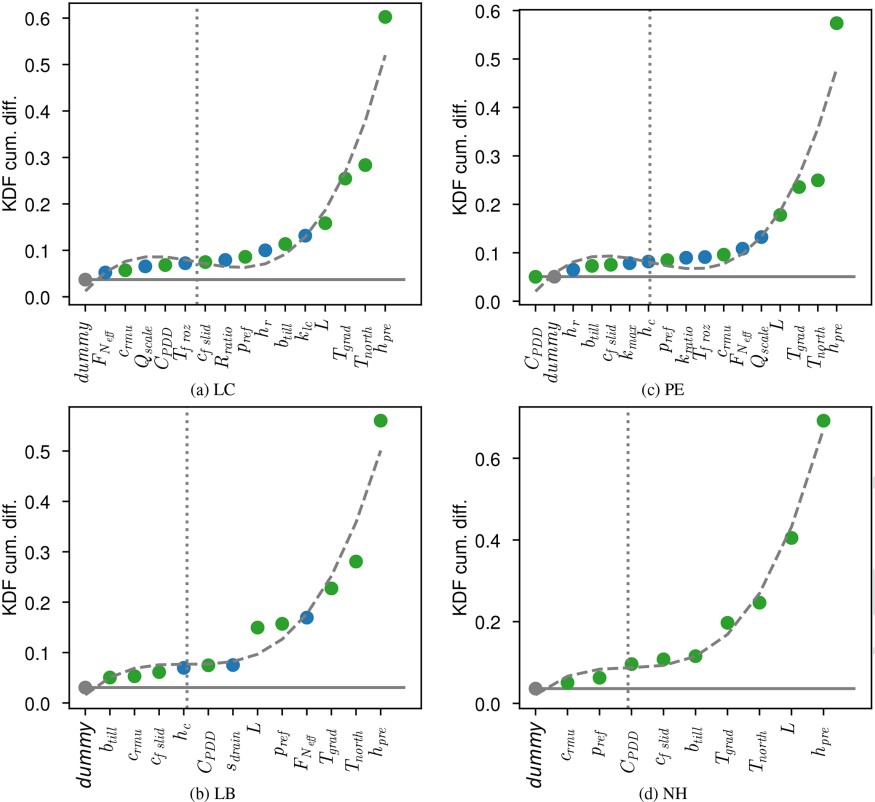

**Figure 5.** Parameters ranked by relative sensitivity by sieving for geometry ($\mathcal{S}^0_{\text{geom}}$, Table 3) relative to the input parameter distribution for each model setup: LC **(a)**, LB **(b)**, PE **(c)**, and NH **(d)**. Blue dots represent subglacial hydrology parameters, green dots the climate parameters, and the gray dot is the dummy parameter. The vertical dotted line indicates the inflection point in the sorted sensitivities used to approximate a transition from diminishingly sensitive to increasingly sensitive parameters. The dashed gray curve shows the fitted third-order polynomial used to calculate the inflection point. The horizontal solid gray line indicates the sensitive threshold of the sensitivity analysis technique given by an unused, random dummy variable.

## 4.13 Surge metric definition

The two most obvious measures of internal oscillation are amplitude and period. This highly non-linear system does not exhibit sinusoidal behaviour, but we can pick surge metrics which approximate these measures. To this end, each surge type was evaluated in two ways – the number of surge events (an indication of periodicity, the number of red dots in Fig. 6) and strength (or speed increase, the height of vertical purple bars in Fig. 6) of surge events (i.e. amplitude).

The background sliding speed of the actual Hudson Strait Ice Stream (HSIS) in the non-surging state is unknown. While this study does not aim to replicate the actual Hudson Strait (HS), we are studying the behaviour of an ice stream and sheet with similar dimensions to the HS and Laurentide ice sheets. As such, labelling and measuring the strength of a surge event needs to be agnostic of quiescent-phase conditions between events. Ice stream acceleration at scales comparable to the HS has not been observed in the modern period. Though significantly smaller than the HSIS and its catchment, the Vavilov ice cap did accelerate from

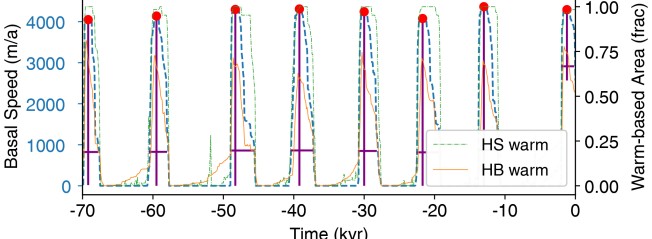

**Figure 6.** Evolution of the ice sheet and idealized Hudson Strait ice stream showing repeated surge events and how metrics are extracted from a sample run. HS basal speed is shown as dashed blue line – which is used to pick surge peaks and estimate prominences – along with the area fraction of warm-based ice within the HS (dash–dotted green line) and its Hudson Bay source region (solid orange line). The red dots show picked event peaks, the vertical purple lines give their "strength" (prominence), and horizontal purple lines show the event duration. HB represents Hudson Bay, and HS represents Hudson Strait CE2

12 to $75\,\mathrm{m\,yr^{-1}}$ between 1998 and 2011 CE (Willis et al., 2018). Satellite observations of the North East Greenland Ice Stream (NEGIS) combine with modelling to show acceleration greater than $1\,\mathrm{m\,yr^{-2}}$ in places between 1985 to 2018 CE (Grinsted et al., 2022). Therefore we define a surge event in this setup as a large increase in spatially averaged HS basal sliding speed ($> 1000\,\mathrm{m\,yr^{-1}}$ over a given 25–100-year acceleration period) over the background, quiescent-phase speed. Velocity can also change during a surge as portions of ice within the HS accelerate over others. Ice stream shear margins can be regions of the fastest velocity changes and ice stream geometry can change over time (Grinsted et al., 2022). As such, we do not define adjacent short-lived changes in velocity as separate surge events.

A typical run with surge events which passes the $\mathcal{S}^0_{\mathrm{geom}}$ sieve is shown in Fig. 6. In order to label surge events (red dots in Fig. 6), we use peak prominence (Virtanen et al., 2020) – drawn from the concept of topographic prominence (height of local maxima above adjacent local minima) – to estimate surge events from the basal velocity time series (1-year sample rate) for each run. This allowed surged metrics to be agnostic of any background value. In order to minimize spurious peaks picked on variations in velocity during a single event, a 401-year median filter was applied. This means that abrupt velocity changes lasting $\sim 200$ years or less will not get picked as events. This is less than the lower bound on HSIS surge duration inferred from ice rafted debris (IRD) by Dowdeswell et al. (1995), who estimate that those surges most likely lasted between 250 and 1250 years on the basis of Heinrich events interpreted in 50 North Atlantic drill cores. A comprehensive review of Heinrich events and IRD age intervals available in the literature by Hemming (Table 3, 2004) infers a mean duration of 495 years, where the lowest estimate is 208 years. The duration for these modelled surge events is calculated as full width at 80 % maximum prominence (height above adjacent local minima).

## 5 HS surging results

The sieves used for sensitivity analysis are shown in Table 3. Sieving the data by ice sheet geometry ($\mathcal{S}^0_{\mathrm{geom}}$) cuts the ensemble size to $\approx 1/6$ to $1/9$: the poro-elastic (PE) system has 3154/19 992 (15.8 %), the linked-cavity (LC) system 3566/18 816 (19.0 %), the leaky-bucket (LB) system 2721/15 288 (17.8 %), and the no-hydrology (NH) ensemble 1382/11 760 (11.8 %) runs. The histograms in Fig. 7 show the frequency of surge events and strength of speedup of those events in the last 50 kyr of each simulation. The lower bound of HSIS surge frequency inferred from the Heinrich Event record (Hemming, 2004; Naafs et al., 2013) is three in 50 kyr. The rate of runs with 3 to 12 surge events in the sieved results is 423/3154 (13.4 %) for PE, 504/3566 (14.1 %) for LC, 836/2721 (30.7 %) for LB, and 75/1382 (5.4 %) for NH. The distribution of the frequency of surge events stem-

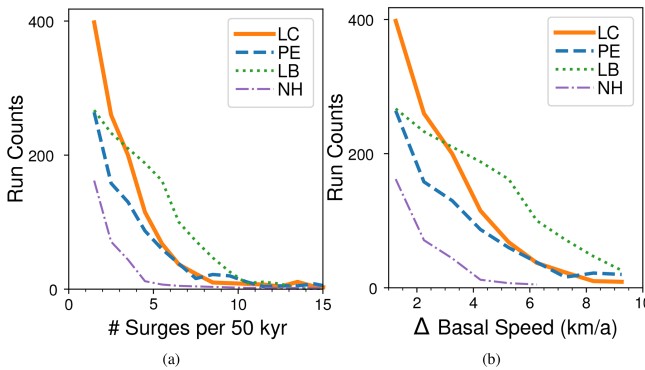

**Figure 7.** Surge event metric distribution across parameterizations by model configuration for runs in the main geometry sieve ($\mathcal{S}^0_{\mathrm{geom}}$, Table 3). The linked-cavity ensemble is shown by the solid orange line, the poro-elastic one by the dashed blue line, the leaky-bucket one by the dotted green line, and the no-hydrology ensemble by the dash–dotted purple line. The number of runs with a given number of surge events in a 50 kyr time frame (referred to here as frequency) is shown in **(a)**. Similarly, the distribution of runs with a given surge strength (peak prominence of spatial mean HS velocity over adjacent local minima) is shown in **(b)**.

ming from each hydrology setup is not significantly different from the others (though LB does have more surges in the 4–7/50 kyr frequency range) nor is the magnitude of the ice stream speedup. The no-hydrology case, however, does differ from those three: the rate of runs with surge events is significantly lower and the frequency and strength of events per run are also lower.

The duration of HS surge events highlights a difference between the three hydrologies: the linked-cavity system yields longer duration events and the trend in duration with increasing event frequency diverges between the linked cavity and the other two hydrology systems. As the duration of surge events necessarily depends on the frequency of those events (having more events in a time period decreases the maximum possible duration of those events), we examine surge duration as a function of the number of events (as shown by the horizontal purple lines in Fig. 6). In Fig. 8 we extract the median surge duration by selecting runs with a given number of events and comparing the duration–frequency trends between the four setups. Frequency levels with 10 or fewer runs passing the sieve are omitted as trends degrade around this level of membership.

As the frequency of surge events in each run increases, the median duration of surges in those runs stays largely flat, perhaps decreasing slightly for both the poro-elastic and leaky-bucket hydrologies. This is not so for the linked-cavity system: the duration of surges increases up to seven surge levels, where it roughly doubles that of the poro-elastic and leaky-bucket hydrologies. This relationship is stronger still when selecting thinner ice sheets with a mean maximum thickness between [2500, 3500] m as shown in Fig. A2. In this geom-

**Table 3.** Sieves used to select runs from each ensemble for analysis.

| Label | Subset | Use |
|---|---|---|
| $\mathcal{S}_{\text{geom}}^{0}$ | $\{H_{\max} \in [3000, 4000]\,\text{m}\} \bigcap \{\Delta y \in [2500, 4000]\,\text{km}\}$ | Main geometry sieve |
| $\mathcal{S}_{\text{geom}}^{\text{high}}$ | $\{H_{\max} \in [3500, 4500]\,\text{m}\} \bigcap \{\Delta y \in [2500, 4000]\,\text{km}\}$ | Additional geometry sieve used in Appendix A to assess thicker ice sheet effect on duration sensitivity |
| $\mathcal{S}_{\text{geom}}^{\text{low}}$ | $\{H_{\max} \in [2500, 3500]\,\text{m}\} \bigcap \{\Delta y \in [2500, 4000]\,\text{km}\}$ | Additional geometry sieve used in Appendix A to assess thinner ice sheet effect on duration sensitivity |
| $\mathcal{S}_{\text{surge}}$ | $\mathcal{S}_{\text{geom}}^{0} \bigcap \{\text{Surge Count} \in [3, 12]\}$ | Sieve used to assess subglacial hydrology parameter contribution to surge frequency |

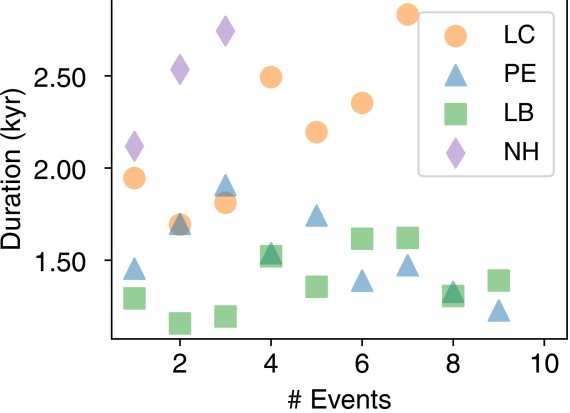

**Figure 8.** This figure shows the trend in surge event duration at different frequencies (number of surge events in a run). The scatter plot shows trends in median duration with an increasing surge frequency. A 10-run minimum was required at each surge frequency level. The no-hydrology setup falls below this requirement after the three-event level, and the linked-cavity setup shows a divergence from the other two subglacial hydrology systems at this point.

etry range, the surge duration decreases at first, reaching a minimum at three surges before steadily increasing in duration until it more than doubles the leaky-bucket surge duration (PE run counts are below the significance threshold). For thicker geometries no differences between the three hydrologies are apparent (Fig. A4).

## 5.1 Sensitivity of surge frequency

Applying a sieve on surge frequency in addition to the geometry sieve ($\mathcal{S}_{\text{surge}}$ in Table 3) highlights the system sensitivity to subglacial hydrology. Figure 8 shows the result from selecting those runs with between 3 and 12 surge events, which is consistent with the minimum number of Hudson Strait surges inferred from the Heinrich Event record and the maximum number of events in the figure. The sensitivity ranking in Fig. 9 is insensitive to whether the sieve upper bound is 8 or 40 events, likely due to the fact that most runs have eight or fewer surge events. For all of the hydrology en-

sembles, the effective pressure normalization exerts the most control on surge frequency (Fig. 9). In the case of the PE and LB ensembles, hydrology parameters give the first- and third-highest sensitivities. $F_{N_{\text{eff}}}$ is highest in both cases, and $k_{\max}$ is third for PE and $h_{\text{c}}$ is third for LB. For LC, the next hydrology parameters do not appear until seventh and eighth place. This may be due to the dual role $F_{N_{\text{eff}}}$ plays in the linked-cavity system: it exerts influence on the sliding velocity, which in turn controls the cavity opening rate which is proportional to effective pressure. In the NH case, soft-bed sliding parameters $c_{\text{rmu}}$ (soft-bed sliding coefficient) and POW$b_{\text{till}}$ (soft-bed sliding law power) are the most important for surge frequency. POWbtill is also the second-most important parameter in both the LC and PE cases.

## 5.2 Relationship between effective pressure and sliding velocity

In Fig. 10, all warm-based points in the ensemble (across the parameter–space–time domain) for each hydrology configuration were cross-plotted in $\log (N_{\text{eff}}) - \log (u_{\text{b}})$ space in order to check for any systematic differences in velocity between the four configurations. If the configurations with subglacial hydrology had increased basal velocities at the ensemble level relative to the no-hydrology case, then the conclusion that subglacial hydrology produces a wider distribution of surge characteristics would provide much less confidence.

The increased incidence of surge behaviour in the hydrology cases is not due to increased sliding – the no-hydrology ensemble exhibits higher basal velocities than the three hydrology ensembles in Fig. 10. This check allowed for an interesting overall comparison between the hydrology configurations. The three hydrology formulations do exhibit differences in $\log (N_{\text{eff}}) - -\log (u_{\text{b}})$ space (Fig. 10). Linked-cavity hydrology produces a bimodal clustering at lower velocities and higher effective pressures and higher velocities and lower effective pressures. This is a stark difference from the other two hydrologies, whose effective pressure distribution simply decays toward lower values. This bifurcation of the effective pressures from a linked-cavity system shows that it

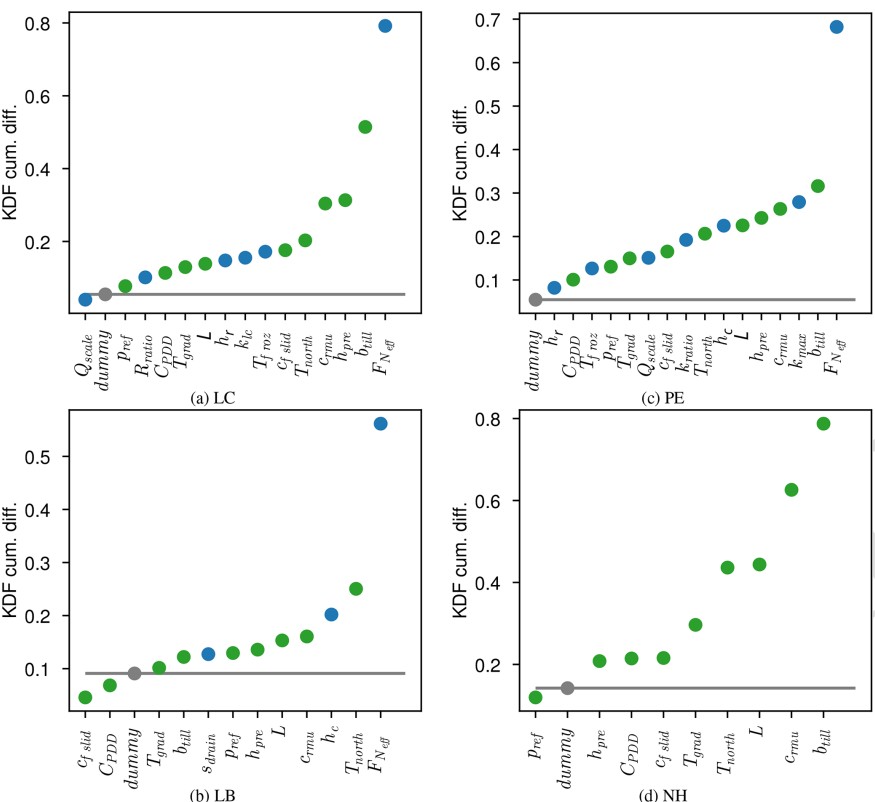

**Figure 9.** Surge frequency sensitivity to model parameters. Parameters are ranked by relative sensitivity by sieving for surge frequencies (3 to 12 events) relative to the geometry sieve ($\mathcal{S}_{\mathrm{surge}}$ relative to $\mathcal{S}^0_{\mathrm{geom}}$, Table 3) for each model setup. The horizontal solid gray line indicates the sensitive threshold of the sensitivity analysis technique given by an unused, random dummy variable. Inflection is weak in each case and so is not used to delineate between sensitive and insensitive parameters (compare Fig. 5).

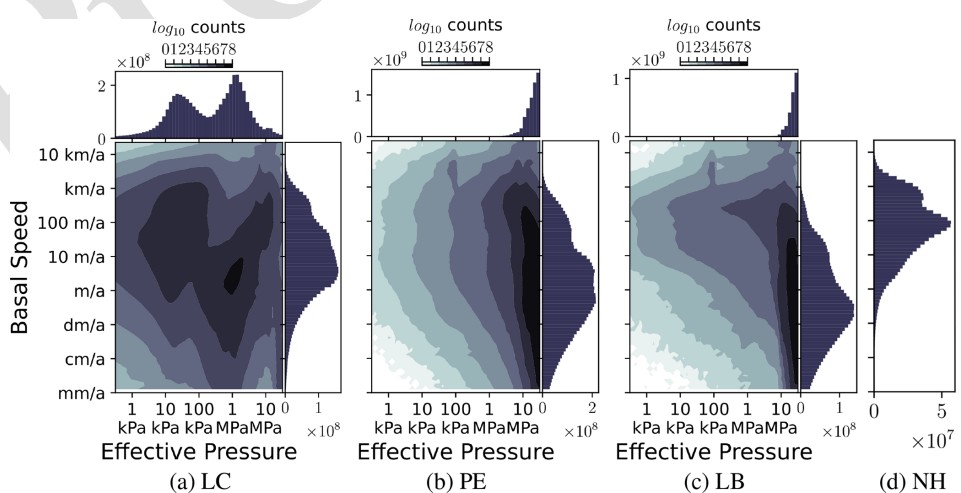

**Figure 10.** Two-dimensional logarithmic histogram of effective pressure and velocity solutions for all warm-based points across the parameter–space–time domain. Fields are output every 100 years. Marginalized distribution for effective pressure and velocity shown along side, sharing the respective axes.

can sustain lower effective pressures than its poro-elastic and leaky-bucket counterparts.

# 6    Discussion of surge contribution

As we show above through sensitivity analysis and ensemble comparison of surge frequency and amplitude, subglacial hydrology is an important process that contributes to the feedbacks which govern Hudson Strait-scale ice stream surging. While the process as a whole matters, the details matter less so – though it does depend on the aspects of ice stream surging under scrutiny. Across the three hydrology setups, the same range of HS basal velocity increase occurs: the magnitude of ice stream speedup is not dependent on the form of the subglacial hydrologic system, and the three models can attain the same velocities within parametric uncertainty. This means that for model experiments looking to realistically capture ice stream surges, a leaky-bucket hydrology (the computationally cheapest of the three) is sufficient. Additionally, the range of frequency of HS surge occurrences is quite similar across the three hydrologies. However, the no-hydrology case falls short of covering the range inferred for actual Heinrich events attributed to HS surging (Naafs et al., 2013). This indicates that the inclusion of some form of coupled subglacial hydrology is important for modelling large-scale surge periodicities on geologic timescales. Once again, however, the exact form of the subglacial hydrology does not matter for the periodicity of the surge onsets.

Plausibly, one might expect that simply increasing the sliding coefficient in the no-hydrology case would generate more surges. We therefore compared the basal velocity distributions between the configurations (Sect. 5.2). The velocity distributions (minimum, mode, maximum) in the hydrology ensembles were slower relative to the no-hydrology configuration in Fig. 10. The range of soft-bed sliding coefficient covered in each ensemble approaches the bounds of plausibility: $c_{\mathrm{rmu}} \in [0.01, 4.0]$, where $c_{\mathrm{rmu}}$ is scaled to give a $3\,\mathrm{km\,yr^{-1}}$ sliding velocity for $30\,\mathrm{kPa}$ basal drag. HS surge behaviour cannot be captured by increasing the sliding coefficient.

Increasing the lapse rate to non-physical bounds can increase the incidence of HS surge events in the no-hydrology case. In the main experiments, the lapse rate is limited to the range $[5, 10]\,^{\circ}\mathrm{C\,km^{-1}}$. However, increasing the lapse rates to $[10, 20]\,^{\circ}\mathrm{C\,km^{-1}}$ increases the rate of surge events. This is because decreasing the surface temperature of ice in the Hudson Bay and Strait both increases the vertical heat diffusion and decreases the temperature of ice advected to the base during a surge event. This enables a stronger thermomechanical surge termination mechanism.

Surge initiation at peak velocity for Hudson Strait-scale ice streams as soon as the pressure melt point is reached is physically implausible. Basal velocity increases after ice becomes warm based and the effective pressure decreases. Inclusion of subglacial hydrology in the coupled system ac-

complishes this. The accommodation of increasing amounts of basal meltwater and pressurization (in the case that channelization does not occur) acts as a system inductance, and the ice stream continues to speed up after becoming warm based. This inductance does not require the lateral transport of meltwater, only the balance of meltwater and a pressure closure dependence on subglacial water thickness.

Though periodicity and strength of surges are similar between the three hydrology-bearing experiments, an interesting distinction occurs when examining the duration of events at varied frequencies. The stabilizing negative feedback of increasing effective pressure at higher basal velocities in the linked-cavity pressure closure gives surge durations longer (up to double the time, depending on frequency) than those of the diagnostic pressure closure of the poro-elastic and leaky-bucket hydrologies. This feedback also results in a bimodal effective pressure distribution (i.e. Fig. 10). When studying ice stream surge behaviour, any of the hydrologies may give the same surge response in terms of frequency and strength of surges. If the study requires a more granular understanding of how long the surge was active, for example when studying the surge timing of multiple ice streams in a catchment (e.g. Payne, 1998; Anandakrishnan and Alley, 1997) or the lifespan of palaeo-ice streams, our results suggest that accounting for the appropriate hydrology system is required.

It is not possible to simulate fully dynamic channelized drainage at the scale studied here; the CFL criterion (Courant et al., 1928) would impose prohibitively long run times for our context. For illustration, Chandler et al. (2013) measured a lower-bound water velocity of $1\,\mathrm{m\,s^{-1}}$ in the channel system. At this speed with our (coarse) resolution of $50\,\mathrm{km}$, a time step of $0.00158$ model years is required. The downgradient routing scheme representing the efficient drainage system is not restricted by CFL, and so the time step depends only on the inefficient system, which is typically in the range of $0.5$ to $0.25$ model years. A dynamic model of the efficient system would increase BrAHMs run time anywhere from a $150$- to a $> 300$-fold rendering simulation of millennial-scale variability infeasible.

Dynamical changes in flow through the efficient system occur on diurnal to seasonal timescales, while the timescales of system features examined here are centennial to millennial. This separation in scale by several orders of magnitude makes it unlikely that dynamical changes in the efficient system (requiring a dynamic model) would be a significant control on the longer-scale variability. However, in a non-linear system, such a control across scales cannot be fully ruled out.

While the treatment of efficient drainage in the model makes it more difficult to closely examine its role in the overall surging system, it is possible to evaluate its role at the ensemble level. At this level it is apparent that efficient drainage does not play a significant role in surging at this scale. Three points bring this to light:

1. The impact on effective pressure from the down-gradient tunnel routing scheme is exaggerated as its modification of the basal water distribution is immediate instead of smooth. This modifies the effective pressure field in both the poro-elastic and linked-cavity systems through the $h_{wb}/h_c$ term in Eq. (4) and the cavity opening rate term in Eq. (8) respectively.

2. The tunnel-switching criterion is well established from a physical mechanistic standpoint (Schoof, 2010). Here we have included a tunnel-switching subgrid uncertainty factor (Sect. 4.8). Sensitivity analysis shows that this parameter, which varies by 3 orders of magnitude, plays little role in surge generation (Fig. 9) where the parameter ranks last in the linked-cavity model and sixth from last in the poro-elastic model. The role of efficient drainage may be greater in the poro-elastic system than the linked-cavity system, suggested by its higher ranking in both the surge sensitivity (Fig. 9) and geometry sensitivity (Fig. 5). This difference in sensitivity may result from the fact that, whereas in the linked-cavity system the rate of change in effective pressure is proportional to the basal water thickness, in the poro-elastic system the effective pressure is directly proportional to the basal water thickness.

3. Though the efficient system is not included in the leaky-bucket configuration, there is little difference in the range of surge frequency and amplitude with respect to the other two systems. The distinction in surge duration stems from the dynamic pressure closure of the linked-cavity system and its direct two-way feedback with sliding velocity.

# 7 Conclusions

The model presented herein passes multiple verification tests and as such is dependable for comparing the effects of structural choices of subglacial hydrology. The sensitivity analysis and ensemble comparison shows that subglacial hydrology is an important control on both ice sheet geometry and on the surging of major ice streams similar in scale to the Hudson Strait Ice Stream. However, depending on the characteristics of interest, the process details do not matter within current parametric uncertainties. The details do not matter for surge periodicity nor strength, but when studying the surge duration the hydrologic details are essential.

Surge behaviours can be produced in the absence of modelling a subglacial hydrology system, but this requires unrealistic assumptions: pushing lapse rates to unrealistic ranges or implementing an un-physical sudden thaw in a large grid cell when the temperature reaches the pressure melt point. Subglacial hydrology provides a system inductance necessary for realistic ice speedup at the temperate transition. The critical components are the accommodation of meltwater and

a meltwater pressure closure, not the mass-conserving meltwater transport itself.

# Appendix A: Surging with thinner and thicker ice sheets

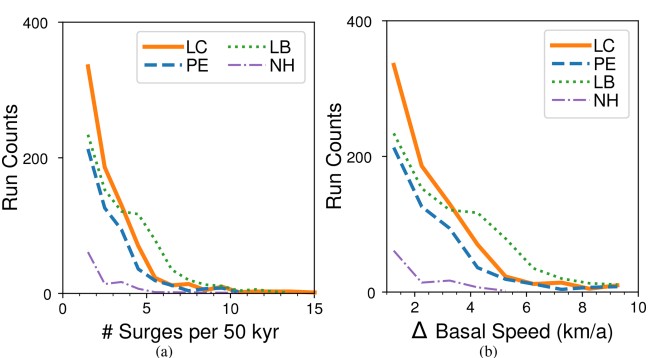

**Figure A1.** Surge event metric distribution across parameterizations by model configuration for runs in the thinner-geometry sieve ($\mathcal{S}_{geom}^{low}$, Table 3). The linked-cavity ensemble is shown by the solid orange line, the poro-elastic one by the dashed blue line, the leaky-bucket one by the dotted green line, and the no-hydrology ensemble by the dash–dotted purple line. The number of runs with a given number of surge events in a 50 kyr time frame (referred to here as frequency) is shown in **(a)**. Similarly, the distribution of runs with a given surge strength (peak prominence of spatial mean HS velocity over adjacent local minima) is shown in **(b)**.

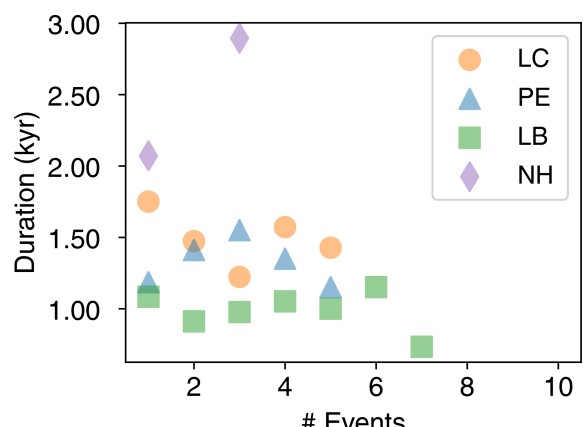

**Figure A2.** Surge event duration at different frequencies for thinner ice sheet sieve ($\mathcal{S}_{geom}^{low}$), [3500, 4500] m. The scatter plot shows trends in median duration with increasing number of surges in a run. The no-hydrology setup falls below this level after the three-event bin.

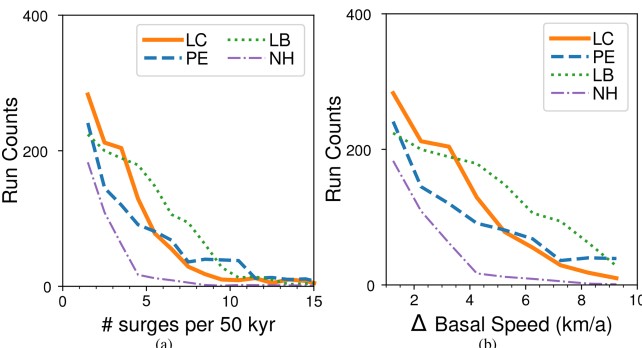

**Figure A3.** Surge event metric distribution across parameterizations by model configuration for runs in the thicker-geometry sieve ($\mathcal{S}_{\text{geom}}^{\text{high}}$, Table 3). The linked-cavity ensemble is shown by the solid orange line, the poro-elastic one by the dashed blue line, the leaky-bucket one by the dotted green line, and the no-hydrology ensemble by the dash–dotted purple line. The number of runs with a given number of surge events in a 50 kyr time frame (referred to here as frequency) is shown in **(a)**. Similarly, the distribution of runs with a given surge strength (peak prominence of spatial mean HS velocity over adjacent local minima) is shown in **(b)**.

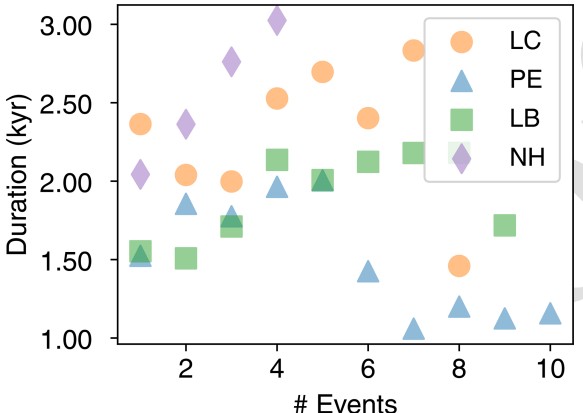

**Figure A4.** Surge event duration at different frequencies for the thicker ice sheet geometry sieve ($\mathcal{S}_{\text{geom}}^{\text{high}}$), [3500, 4500] m. This figure shows the trend in surge event duration at different frequencies (number of surge events in a run). The scatter plot shows trends in median duration with an increasing surge frequency. A 10-run minimum was required at each surge frequency level.

## Appendix B: Subglacial hydrology model solver

BrAHMs2.0 solves the conservative transport equation for the distribution of subglacial water (Eq. 3) and the effective pressure evolution equation (Eq. 8) using combined explicit and semi-implicit methods. Time integration is done first with Heun's method for the initial time step followed by a leapfrog trapezoidal predictor–corrector method (Kavanagh and Tarasov, 2018). To avoid time splitting, Heun's method is called after every 10 leapfrog steps (varying the number of leapfrog steps had little effect on the solution in tests).

The verification of this scheme and its implementation is presented in Appendix D with a four-pronged approach. The model is shown to give spatially symmetric solutions given symmetric boundary conditions. The convergence is examined for the spatial and temporal discretizations and found to approximately match the expected rate for each scheme: the first-order upstream finite-volume implementation spatially converges at a linear rate, while the second-order leapfrog trapezoidal implementation temporally converges nearly quadratically. The associated partial differential equations, however, are non-linear, coupled, and likely to have non-local responses. As such, assessing the expected convergence rate of this system is not straightforward (Tadmor, 2012). In Appendix D the model is shown to also conserve mass and match the solution of another numerical model with similar physics (Werder et al., 2013).

## Appendix C: Subglacial hydrology model assumptions

The physics of the linked-cavity system are highly non-linear. As such, a set of simplifying assumptions is required to make numerical modelling of this framework feasible.

1. Wall melting is not a control on cavity size until tunnels are opened. Drainage systems switch from inefficient to efficient for a given value of flux. Schoof (2010) showed that the evolution of the subglacial drainage system (described in Eq. 5) gives a bifurcation between cavity style and tunnel style drainage networks. Given effective pressure, the cavity opening speed is dominated by basal sliding below a certain flux and by runaway wall melting above it.

2. At timescales of continental-scale ice sheets, tunnels drain water instantaneously. The timescale of drainage through subglacial tunnels is less than a single melt season, much shorter than the centennial to millennial-scale changes this model is applied to. This assumption alleviates CFL violations from fast tunnel flux, which would render modelling on the long timescales of glacial cycles infeasible.

3. Cavities are filled with water. Consider the timescale for the closure of a recently drained cavity given various combinations of ice sheet overburden (thickness, m) and sliding velocities. This timescale for closure (from Eq. 6) is given by

$$T = \frac{S}{u_{\text{b}} h_{\text{r}} - c_2 N^3 S}. \tag{C1}$$

The range of timescales, assuming speed in the range of 1–1000 m yr$^{-1}$ and ice overburden thickness greater than 200 m, is shown in Fig. C1, where the maximum time for closure is around 2 weeks, less than the minimum time step of 0.125 years in the hydrology model.

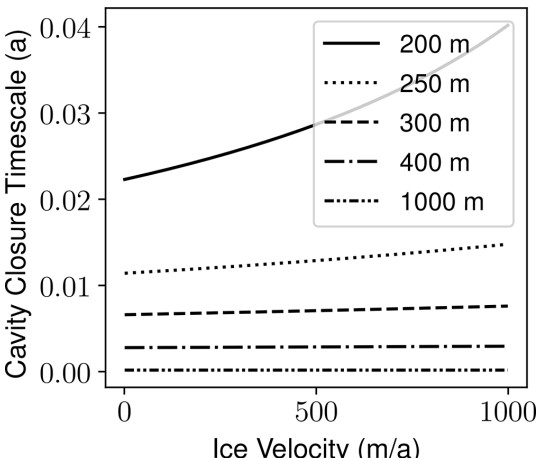

**Figure C1.** Cavity closure times at varied ice sheet thickness and sliding speeds.

## Appendix D: Subglacial hydrology model verification

Oreskes et al. (1994) describe model verification in general as the task of demonstrating model veracity, correctly asserting that no model can ever be proven – only disproven. However, this problem is not unique to computational model testing, this is a more philosophical epistemological problem. As Sornette et al. (2007) identifies, we do not prove models, we simply build our trust in them through a series of failed attempts to disprove them. In this section, we document performance on some simple tests which every model should pass before any amount of confidence can be conferred.

Following others (e.g. Sornette et al., 2007), we take model verification to be more pedestrian than validation: a test that the computational model actually solves the model equations as intended – or, as Roache (1997) defines it, "solving the equations right." Meanwhile, we take validation as the converse of Roache (1997), i.e. "solving the right equations." Validation-wise, in this work we are showing not that the right equations were solved but that this seems to be of low consequence.

The results presented in this section were reached in an effort to expose errors in the models, the lowest-hanging fruit in gaining confidence in the model solutions. The verification strategy in this section is to satisfy the following:

1. model solutions are symmetric given symmetric input,

2. model solutions converge under increasing spatial and temporal resolution,

3. mass is conserved, and

4. models using similar physics should have similar solutions.

Using simplified setups, expected behaviours are straightforward and in some cases may be calculated by hand

(though hand calculations are not shown here). By using a progression of most simple to increasingly complex model setups for testing, model behaviour can be verified against expected behaviour and shown capable of simulating increasingly realistic environments. Here we demonstrate that the model correctly solves the equations. A progression of forcings and couplings were used, of which the transient, two-way coupled solutions from the least stable parameters (while still physical) are shown.

Parabolic surface topographies haven been used to approximate non-streaming ice sheet topographies (e.g. Mathews, 1974). The Subglacial Hydrology Model Intercomparison Project (SHMIP) (de Fleurian et al., 2018) uses such an ice sheet surface (depicted in Fig. D3) and provides solutions to models using similar physics as the model herein. This therefore provides an appropriate test bed. This SQRT_TOPO surface is given by

$$z_s = 6.0 \left( \sqrt{x + 5000} - \sqrt{5000} \right) + 10, \tag{D1}$$

and the flat base $z_b = 0$.

Testing of the linked-cavity system with a Darcy–Weisbach flux model configuration (Eqs. 8 and 1) is presented here as this is the most non-linear form and a new addition to the model.

The basal sliding velocity is determined by the effective pressure from Eq. (8):

$$u = k_{\text{slide}} \frac{\tau_b}{N_{\text{eff}}}, \tag{D2}$$

where $k_{\text{slide}} = 5.0 \times 10^1 \, \text{m s}^{-1}$ is a scaling constant, an effective pressure regularization $10 \, \text{Pa}$ is applied for numerical stability, and basal shear stress ($\tau_b$) is calculated from the constant driving stress ($\tau_d$):

$$\tau_b = \tau_d = \rho_{\text{ice}} g H \frac{\partial H}{\partial x}. \tag{D3}$$

### D1 Symmetry test

Spatial symmetry at each spatial resolution was calculated as the sum of the difference between the two ice sheet halves across the divide. This difference is 0 for all fields showing perfect symmetry.

### D2 Temporal resolution test

Here we test the effect of changing the length of the time step in the basal hydrology on model solution using the SHMIP SQRT_TOPO setup (depicted in Fig. D3). The per-run discrepancy with respect to the shortest time step shown in Fig. D1 is calculated as

$$\text{ERR}(\Delta t_i) = \sum_k^{N_y} \sum_j^{N_x} \left| N_{\text{eff}}^{jk}(\Delta t_i) - N_{\text{eff}}^{jk}(\Delta t_{-1}) \right|. \tag{D4}$$

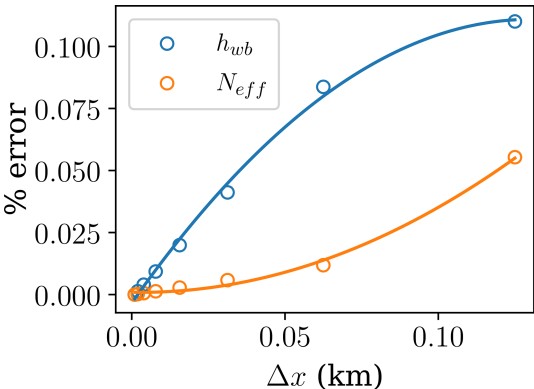

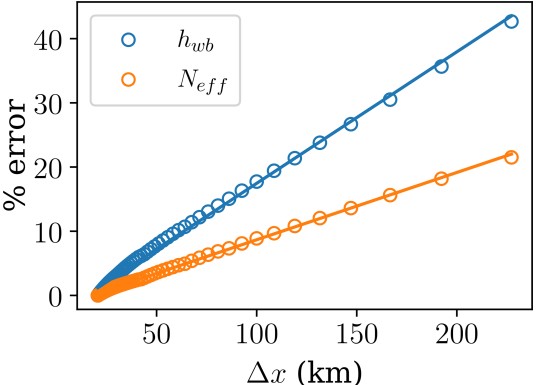

**Figure D1.** Convergence with decreasing time step. Each field is normalized with the normalization factor shown in the legend (maximum). The points are fitted with a degree-2 polynomial to show the approximately quadratic rate of convergence.

**Figure D2.** Difference in mean flowline solutions for unsteady SHMIP square root ice sheet topography at increasing spatial resolution, at end of 10 kyr run. The points are fitted with a line to demonstrate the match with the order of the numerical scheme.

As a first test of convergence under increasing temporal resolution (decreasing time step length), the hydrology model was run to steady state under SHMIP scenario A (constant $2.5\,\mathrm{mm\,yr^{-1}}$). Seeing convergence at shorter time steps for the steady forcing, an unsteady sinusoidal meltwater forcing was applied (50-year period, $3.5\,\mathrm{mm\,yr^{-1}}$ amplitude). The convergence for the unsteady case is shown in Fig. D1 with the error metric of Eq. (D4). The rate of convergence is approximately quadratic as expected for the $\mathcal{O}\left(\Delta t^2\right)$ leapfrog trapezoidal scheme.

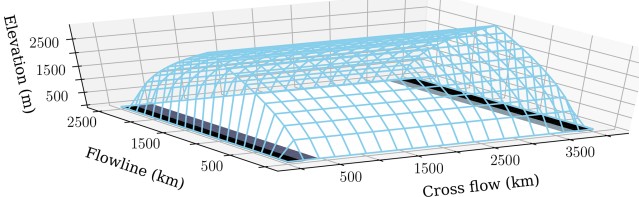

**Figure D3.** Ice sheet configuration used in SHMIP with basal temperature (black $= -40$; white $= 0.01$).

### D3 Spatial resolution test

Here we show the effect of varying spatial resolution on the model solution. The model was run to steady state with prescribed melt and basal velocity ($1.75\,\mathrm{m\,yr^{-1}}$ and $2.0\,\mathrm{m\,yr^{-1}}$ respectively). For this test, the SHMIP setup was used as shown in Fig. D3. The SQRT_TOPO flowline length from divide to toe was set to 2500 km, and the number of grid cells was adjusted, so that the highest resolution was $\Delta x_i = 22.66\,\mathrm{km}$: $\{n_i = 2i + 1\}$ for $i \in [11, 121]$, $i \in \mathbb{N}$ and $\Delta x_i = 2500\,\mathrm{km}/n_i$. The model solution at each resolution was linearly interpolated to the highest-resolution grid, and the sum of the element-wise difference with the highest resolution was used for the (L1 norm) error metric, in keeping with Eq. (D4):

$$\mathrm{ERR}\left(\Delta x_i\right) = \sum_k^{N_y} \sum_j^{N_x} \left| \Lambda^{jk}\left(\Delta x_i\right) - \Lambda^{jk}\left(\Delta x_{-1}\right) \right|. \tag{D5}$$

Figure D2 shows the convergence of model solutions (same set as Sect. D2) at increasing spatial resolution (shorter cell width). The numerical order of the upwind and finite-volume schemes used here is $\mathcal{O}\left(\Delta x\right)$. The approximately linear rate of convergence in Fig. D2 matches this numerical order.

### D4 Mass conservation

Mass conservation is demonstrated by comparing flux at the margin to source rates of water or sediment within the ice sheet: the integral of the melt rate over the ice sheet less the total flux through the margin will give the change in basal water volume over time. Integrating this change up to each time step will give the basal water volume at each time step, which can be compared to model-calculated basal water volume in order to assess mass conservation.

To test mass conservation with unsteady input, we applied a sinusoidal meltwater forcing,

$$m_\mathrm{t}^{jk} = \frac{\mathrm{melt}}{2} \sin \frac{2\pi}{T} t + \frac{\mathrm{melt}}{2} + \frac{\mathrm{melt}}{4} \sin 12 \frac{2\pi}{T} t + \frac{\mathrm{melt}}{4}$$
$$+ \frac{\mathrm{melt}}{8} \sin 25 \frac{2\pi}{T} t + \frac{\mathrm{melt}}{8}, \tag{D6}$$

(where $T = 50\,\mathrm{kyr}$ is the longest and highest-amplitude period and $\mathrm{melt} = 3.5\,\mathrm{mm\,yr^{-1}}$), to the SQRT_TOPO setup (depicted in Fig. D3) and calculated basal sliding velocity dynamically as in Eq. (D2). Here we assume incompressibility of water such that volume is scaled mass.

A net volume of basal water time series was calculated by time-integrating the net of input and output, $\mathrm{net}_\mathrm{hyd}^{t_i}$, up to

each time step $t_i$:

$$\text{net}_{\text{hyd}}^{t_i} = \int\limits_0^{t_i} \left( \int\limits_{\mathcal{A}} m_{\text{t}} \mathrm{d}a - \oint\limits_{\mathcal{S}} \boldsymbol{Q} \cdot \hat{\boldsymbol{n}} \mathrm{d}S \right) \mathrm{d}\tau, \qquad (\text{D7})$$

where $\mathcal{A}$ is the area covered by ice, $m_{\text{t}}$ is the melt at the ice sheet base (Eq. D6), $\mathcal{S}$ is the ice margin (interface beyond which ice thickness is 0), and $\boldsymbol{Q} \cdot \hat{\boldsymbol{n}}$ is the flux through the margin, The $\text{net}_{\text{hyd}}^{t_i}$ time series was then compared against modelled total water volume ($V_{\text{hyd}}^{t_i}$) to calculate mass conservation error ($\text{ERR}^{t_i}$):

$$\text{ERR}_{\text{hyd}}^{t_i} = \frac{|\text{net}_{\text{hyd}}^{t_i} - V_{\text{hyd}}^{t_i}|}{V_{\text{hyd}}^{t_i}}, \qquad (\text{D8})$$

where $V_{\text{hyd}}$ is the volume of water under the ice sheet.

The dynamic model outputs from this test are summarized in Fig. D4. This mass conservation test shows a maximum error of 0.052 % between the model output and the calculation in Eq. (D7) (given in Eq. D8).

## D5 Comparison with Werder et al. (2013) results for SHMIP

Results for this model are compared with output of the Glacier Drainage System model (GlaDS, Werder et al., 2013) employing the same physics: a continuum representation of a linked-cavity system with Darcy–Weisbach flux shown in Fig. D5. While the model of Werder et al. (2013) is similar to this one, there are noteworthy differences. Werder et al. (2013) use an unstructured mesh and finite-element discretization, and the channel elements are always active (with water exchanged between the channels at the edges and the distributed system at the subdomains). This is in contrast to BrAHMs2.0, in which the channel system switches on in a particular cell given a flux criterion (and uses finite-volume discretization with a regular Cartesian grid). We therefore use the SHMIP scenario in which the least amount of channelized flux is active in order to get the most structurally consistent comparison between the two models. BrAHMs2.0 closely reproduces the flux and effective pressure solutions for this scenario, concluding our verification that we solved the equations "right".

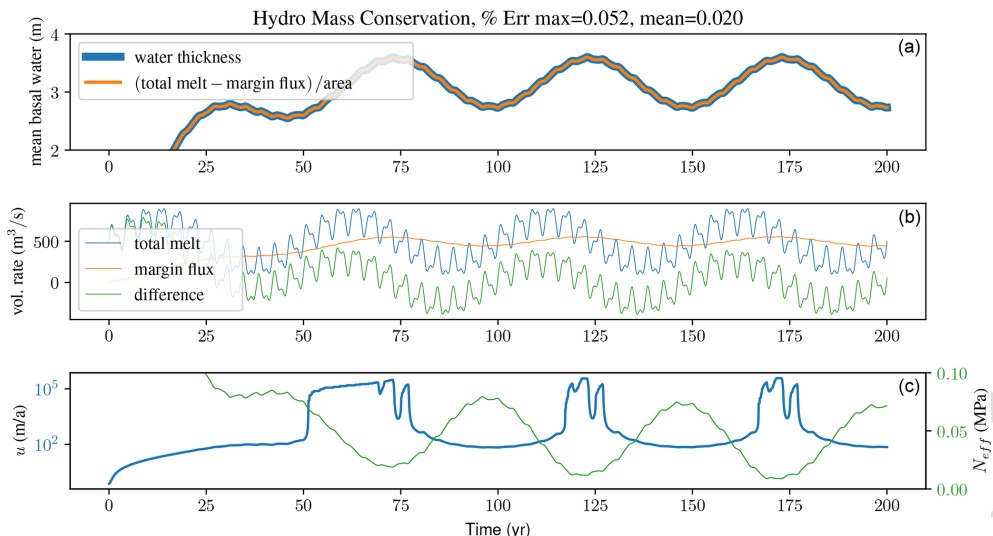

**Figure D4.** Assessment of mass conservation for subglacial hydrology model given steady square root ice sheet topography, flat basal topography, and sinusoidal ice sheet basal meltwater generation ($m\,yr^{-1}$) given in Eq. (D6). The basal sliding velocity is calculated from driving stress and effective pressure (Eq. D3) over a 200-year modelled time period. The model solution for basal water thickness is compared with the time-integrated difference between basal melt and flux out of the margin (Eq. D7) in panel **(a)** (near-complete visual overlap). For an illustration of model input and response, panel **(b)** shows the basal meltwater, flux out of the margin, and the difference between the two over time. Panel **(c)** shows dynamically calculated, two-way coupled basal velocity in blue and effective pressure in green.

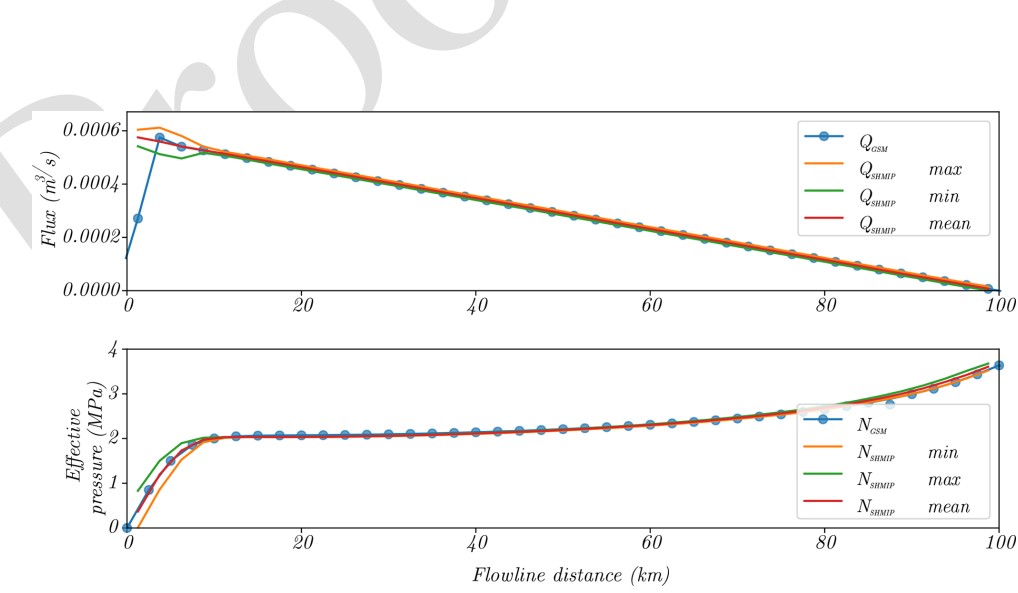

**Figure D5.** Comparison of our model solution with the SHMIP tuning set, which used output from the model of Werder et al. (2013), which uses similar physics to BrAHMs2.0 in the linked-cavity configuration.

## Appendix E: Discretization

### E1 Pressure closure of Bueler and van Pelt (2015)

Here we use the time-varying water pressure calculation of Bueler and van Pelt (2015). The rationale summarized here is shown by Bueler (2014). Here the subglacial and englacial hydrologic systems are assumed to be in perfect communication and their co-evolution is described. The englacial hydrologic system is analogized to a rigid "pore space" (comprised of crevasses, moulins, englacial channels, and inter-granular porosity). The total volume of water is the sum of *eng*lacial and *sub*glacial water:

$$V_{\text{tot}} = V_{\text{eng}} + V_{\text{sub}}, \tag{E1}$$

and the mass balance for incompressible water is

$$\frac{\partial V_{\text{tot}}}{\partial t} = -Q_{\text{out}} + Q_{\text{in}} + \frac{m}{\rho_{\text{w}}} \tag{E2}$$

from total flux in and out of a control section of the system plus any sources (volume water, $\frac{m}{\rho_{\text{w}}}$) within that section. This section is of the area $\Delta x$ by $\Delta y$, and pressure in the connected englacial-subglacial system is given by the hydrostatic head in the englacial part:

$$P_{\text{w}} = \frac{\rho_{\text{w}} g}{\Delta x \, \Delta y \phi_{\text{eng}}} V_{\text{eng}}. \tag{E3}$$

The effective englacial porosity (ice volume relative proportion of connected englacial void space) is $\phi_{\text{eng}}$. Cavity volume within an area of bed with roughness wavelength $l_{\text{r}}$ (cavity-generating obstacle spacing) is

$$V_{\text{sub}} = n_{\text{cav}} V_{\text{cav}} = \frac{\Delta x \, \Delta y}{l_{\text{r}}^2} V_{\text{cav}}, \tag{E4}$$

where $n_{\text{cav}}$ is the number of cavities in the given bed section and $V_{\text{cav}}$ is their average volume. Differentiating this gives the change in pressure with time:

$$\begin{aligned}
\frac{\partial P_{\text{w}}}{\partial t} &= \frac{\rho_{\text{w}} g}{\Delta x \, \Delta y \phi_{\text{eng}}} \frac{\partial V_{\text{eng}}}{\partial t} \\
&= \frac{\rho_{\text{w}} g}{\Delta x \, \Delta y \phi_{\text{eng}}} \frac{\partial V_{\text{tot}} - V_{\text{sub}}}{\partial t} \\
&= \frac{\rho_{\text{w}} g}{\Delta x \, \Delta y \phi_{\text{eng}}} \left\{ Q_{\text{in}} - Q_{\text{out}} + \frac{m}{\rho_{\text{w}}} - \frac{\Delta x \, \Delta y}{l_{\text{r}}^2} \frac{\partial V_{\text{cav}}}{\partial t} \right\}.
\end{aligned}$$

The $\left( 1/l_{\text{r}}^2 \right) \frac{\partial V_{\text{cav}}}{\partial t} = \frac{\partial h_{\text{cav}}}{\partial t}$ derivative is given by the opening and closing balance in Eq. (7):

$$\begin{aligned}
\frac{\partial P_{\text{w}}}{\partial t} = \frac{\rho_{\text{w}} g}{\phi_{\text{eng}}} &\left\{ \frac{Q_{\text{in}} - Q_{\text{out}} + \frac{m}{\rho_{\text{w}}}}{\Delta x \, \Delta y} \right. \\
&\left. -u_{\text{b}} \left( h_{\text{r}} - h_{\text{wb}} \right) / l_{\text{r}} + c_2 [P_{\text{ice}} - P_{\text{w}}]^n \right\}.
\end{aligned}$$

Here opening due to wall melting has been omitted (see assumption 1) relative to what is shown by Bueler (2014). As $\Delta x \to 0$ and $\Delta y \to 0$, the difference between the fluxes in versus out of the control section goes to the divergence of the fluxes within it.

$$\begin{aligned}
\frac{\partial P_{\text{w}}}{\partial t} = \frac{\rho_{\text{w}} g}{\phi_{\text{eng}}} &\{ -\mathbf{\nabla} \cdot \mathbf{Q} + m_{\text{t}} \\
&-u_{\text{b}} \left( h_{\text{r}} - h_{\text{wb}} \right) / l_{\text{r}} - c_2 [P_{\text{ice}} - P_{\text{w}}]^n \},
\end{aligned} \tag{E5}$$

with $m_{\text{t}}$ the source of water in thickness per unit time. We assume that water only travels laterally through the subglacial system, and so all fluxes are through the linked cavities.

*Code and data availability.* Analysis and data are available at Zenodo (https://doi.org/10.5281/zenodo.10018301, Drew, 2023).

*Author contributions.* MD performed subglacial hydrology model development, experimental design and execution, analysis, and writing. LT maintains the GSM, advised on experimental design, and edited the paper.

*Competing interests.* The contact author has declared that neither of the authors has any competing interests.

ther geographical representation in this paper. While Copernicus Publications makes every effort to include appropriate place names, the final responsibility lies with the authors.

*Acknowledgements.* The figures in this work were generated with Matplotlib (Hunter, 2007). Computational resources were provided by the Canadian Foundation for Innovation (CFI) and ACENET – a regional partner of the Digital Research Alliance of Canada. We thank the two anonymous reviewers for their helpful suggestions.

*Financial support.* This work was supported by NSERC Discovery (grant no. RGPIN-2018-06658), the Canadian Foundation for Innovation, and the German Federal Ministry of Education and Research initiative for Research for Sustainability through the PalMod project.

*Review statement.* This paper was edited by Alexander Robinson and reviewed by two anonymous referees.

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

**Remarks from the language copy-editor**

CE1    Please confirm the changes.

CE2    Please confirm the changes.

**Remarks from the typesetter**

TS1    Please let us know where Table 2 should be mentioned in the text.