# Peer review of "Surging of a Hudson Strait Scale Ice Stream: Subglacial hydrology matters but the process details mostly don't"

_The Cryosphere, 2022_

## Referee Comment (RC1)

**Review of the article entitled "Surging of a Hudson Strait Scale Ice Stream: Subglacial hydrology matters but the process details don't" by M. Drew and L. Tarasov**

**1 General comments**

This study investigates the implications of introducing different subglacial hydrology model formulation for the surge dynamics of an Hudson Strait like ice stream. The manuscript first introduces the different approaches that are used for the subglacial hydrology modelling before providing a verification of theses implementations. The authors then present a framework for long term simulations of an Hudson Strait like experimental design that allows to perform a large ensemble of simulations on long time scales. The results of the ensemble is then presented both to emphasise the parameter sensitivity of the different hydrology model and the differences that arise between the different approaches.

I found this study to be overall well presented and clear on an interesting and topic that as not been studied in details before. In an effort to give a clear description of their approach and models, the authors end up with a quite lengthy paper but I expect that it can not really be avoided here, I only noted a few places where some paragraphs may be omitted which would make room for more critical information.

As a general note, I am under the impression that some of the conclusion of the study and particularly the one relative to the unimportance of "process details" is overstated. I agree that the results shown here on the Hudson Strait surges frequencies seems to be unchanged by the different hydrology modelling approach used here but it feels to me that it is a bit thin to have such a strong conclusion.

Regarding the hydrology processes themselves, I am unsure if the efficient drainage is applied to the linked cavity system only or two both the mass transporting hydrology setups. I also am not sure why this part of the model is not compared to the GlaDS model, to me the shift between efficient and inefficient drainage is one of the crucial point of a subglacial hydrology model and I would be very interested to see how the approaches used here compare to GlaDS with a higher recharge scenario (like A5 from SHMIP).

The presentation of the ensemble results and the sieve treatment made this part easy to read and interpret and that was a very interesting read, I noted a few places where referencing could be improved. I would also like to suggest a few changes to the figures that are listed in the minor comments bellow and would improve readability.

**2  Specific comments**

Bellow are some comments on specific sections of the manuscript.

Introduction. The reference to Hank et al. here is a bit problematic as it is not clear to me what we are talking about, perhaps a succinct description of the method developed in the Hank paper should be introduced here.

Section 2. I find the description of the inefficient drainage systems to be well presented here but I am less convinced by the description for the efficient drainage system, I am not sure for example if the efficient drainage applies to both poro-elastic and linked cavity systems or to only one of those.

Section 3. The sub-sectioning of this part could be improved, perhaps the very short introduction can be replaced by the current sub-section 3.2 which would serve the purpose of introduction here. I would also suggest to have a specific section for the hydrology model verification which in my opinion is an important part of the study. The assumption of the hydrology model are also quite important in the description of the model and should in my opinion appear in the section 3 rather than in the supplementary. In order to shorten the main manuscript, section 3.1.1 could be moved to the supplementary materials. Regarding the subsection on model validation, I am unsure of the necessity of the introductory paragraph I would however like to see a comparison of the model to GlaDS with a higher input scenario to assess the changes that are due to this part of the model.

Section 4. Part of this section might be reduced, I feel that the parameters for which only a few lines are given with no strong argumentation could only appear in a table with a reference cited to justify the range. I am not sure of the temperature description that is given in equation 18. Following the equation given there and the map of LISsq the temperature at the northern end of the domain would not be $T_{north}$ but $T_{north} + 5000 T_{grad}$ also, the elevation dependant term is summed with the others but the lapse rate is positive which would give higher temperature at higher elevation.

Discussion. As for the title I feel that the discussion here is a bit to strong in term of wording. I agree that the details of hydrology approaches presented here do not impact greatly the frequency of surges but they may be important for other applications (as shown

in figure 13 for example). It feels also necessary here to underline the shortcomings of the present study and their potential implications on the conclusion.

Here are more technicla comments given with their line number.

- Line 7: Is "BraHms" here an acronym, if yes it should be spelled.

- Line 8: I am not sure what "systems" are referred to here.

- Line 19: I would not say that the role of subglacial hydrology at time scales larger than a season is clear, so perhaps this sentence can be rephrased.

- Line 21: The sentence ending on this line is not the clearest and could be reworded.

- Line 32: Studies by Fürst et al. (2015); Shannon et al. (2013) might be relevant here and could be added.

- Line 43: "begin to rapidly slide" "to is missing.

- Line 48: References for the hydrology formulations should be added.

- Line 52: A reference is missing for GSM.

- Line 64: I expect that "HS" stands for Hudson Strait but it has not been introduced before.

- Line 70: It should be "or" in place of "and".

- Line 78: I am not sure why porous media is used here where poro-elastic is used everywhere else.

- Line 79: Flowers (2015) should be added here.

- Line 91: Smith et al. (2021) could be added here.

- Line 94: I find this formulation unclear.

- Line 116: I feel that the formulation here could be shortened giving only the new equations are the exponents of equation 1.

- Line 121: $P_{water}$ is already defined above.

- Line 133: I feel that the formulation here could be shortened giving only the new equations are the exponents of equation 1.

- Line 133: $k$ is referred to here bu $K$ is in the equation.

- Line 135: $N_{eff}$ is usually referred to as effective pressure, is there a reason for it being pressure closure here?

- Line 141: $N_{eff}$ was defined above already.

- Line 146: There is a missing reference here.

- Table 1: The acronyms of the table should be described in the caption.

- Line 174: Numbering is missing for those equations and it should be $h_{cav}$ in place of $h$.

- Equation 9: $phi_{eng}$ and $m_t$ are not described.

- Line 182: $_{drain}$ and $s_{melt}$ could be added on this line for description.

- Line 183: A comma is missing in the reference.

- Line 198: "v2" is missing for BraHms.

- Line 245: The sentence starting on this line should be rephrased.

- Line 258: Figure 1 shows the convergence for all parameters not only $N_{eff}$ so I am not sure here what *Lambda* stands for.

- Line 307: "Bay/Strait" is capitalised here but not bellow.

- Line 307: "HEINO" here needs a reference.

- Table 2: This table is missing from my version of the manuscript.

- Equation 20: There is no description of the parameters of this equation.

- Line 350: $k_{min}$ value should be fixed.

- Line 350: Is there a reason to have upper case and lower case "k" for conductivities.

- Figure 5: Colours on this figure should be changed for more colourblind friendly versions.

- Figure 5: I am not sure which Flow speed is referred to on the x axis.

- Equation 21: Operator log and arctan should not be slanted fonts.

- Equation 21: is there a reason for the renaming of $h_c$ to h_crit_dpe here?

- Line 358: I have not seen $T_{bp}$ used anywhere else.

- Equation 22: I have not seen $K_f$ defined anywhere.

- Equation 22: $T_{froz}$ here and in the text have different font.

- Line 371: "e.g." should be before the reference.

- Line 372: I am not sure that the precision on the referencing are needed here.

- Line 385: I am not sure of the meaning of "communition".

- Line 390: See my comment above on the clarification for the Hank paper.

- Line 397: I am not sure of the point of the sentence starting on this line.

- Line 402: I think that the reference to fig. 6 here is misplaced.

- Figure 6a: Label is missing on the y axis.

- Line 452: Shouldn't reference here be to figure 8 rather than 12.

- Line 457: Shouldn't reference here be to figure 8 rather than 12.

- Figure 8: The horizontal line is not described in the caption.

- Figure 9: Line colours should be given in the caption, the legend should be moved for clarity and upper case letter used as in the caption.

- Line 476: Reference to figure 9 should be added here.

- Line 489: Reference to figure 9 should be added here.

- Line 497: I am not sure what "tbl. 3" references to is it in the referenced study?

- Line 504: "on of HSIS" either "on" or "of" should be removed.

- Figure 10: The colours between PE and NH should be changed for colourblind friendliness.

- Figure 10: On this type of figures perhaps a line would be clearer than the histogram to compare the different models.

- Figure 10: "Run Density" of panel a is missing ticks and labels.

- Line 527: "Fig 12" should be referenced here.

- Figure 11: The colours between PE and NH should be changed for colourblind friendliness.

- Figure 11: Upper panel y-axis of panel a is missing a label.

- Line 535: A reference to figure 13 should be added here.

- Figure 12: The horizontal line is not described in the caption.

- Figure 12: I wonder why the delineation between sensitive and insensitive is not presented as in Figure 8.

- Figure 13: The font on this figure is a bit small.

- Line 551: The paragraph starting on this line could be clearer.

- Line 554: The number of the referenced figure is missing.

- Figure A1 and A3: Same colour issue as before, and the grey region is not described in the caption, ticks and labels on the left panel are missing.

- Figure A2 and A4: Same issues as figure 11.

- Figure C1: The colours should be changed for colourblind friendliness.

- Line 634: The number of the equation referenced here is missing.

**References**

Flowers, G. E. (2015). Modelling water flow under glaciers and ice sheets. *Proc. R. Soc. A*, 471(2176):1–41.

Fürst, J. J., Goelzer, H., and Huybrechts, P. (2015). Ice-dynamic projections of the Greenland ice sheet in response to atmospheric and oceanic warming. *Cryosphere*, 9(3):1039–1062.

Shannon, S. R., Payne, A. J., Bartholomew, I. D., van den Broeke, M. R., Edwards, T. L., Fettweis, X., Gagliardini, O., Gillet-Chaulet, F., Goelzer, H., Hoffman, M. J., Huybrechts, P., Mair, D. W. F., Nienow, P. W., Perego, M., Price, S. F., Smeets, C. J. P. P., Sole, A. J., van de Wal, R. S. W., and Zwinger, T. (2013). Enhanced basal lubrication and the contribution of the greenland ice sheet to future sea-level rise. *Proceedings of the National Academy of Sciences*, 110(35):14156–14161.

Smith, A. M., Anker, P. G. D., Nicholls, K. W., Makinson, K., Murray, T., Rios-Costas, S., Brisbourne, A. M., Hodgson, D. A., Schlegel, R., Anandakrishnan, S., and et al. (2021). Ice stream subglacial access for ice-sheet history and fast ice flow: the beamish project on rutford ice stream, west antarctica and initial results on basal conditions. *Annals of Glaciology*, 62(85-86):203–211.

---

## Author Comment (AC1)

**Response to Reviewer 1 of article entitled "Surging of a Hudson Strait Scale Ice Stream: Subglacial hydrology matters but the process details don't"**

Matthew Drew[1] and Lev Tarasov[1]

[1]Department of Physics & Physical Oceanography, Memorial University of Newfoundland, St John's, Canada

**Correspondence:** Matthew Drew (mdrew@mun.ca)

Reviewer 1 has identified a number of improvements to this manuscript and we are grateful to their detailed examination and thoughtful suggestions. We agree that the paper is rather long. However, like Reviewer 1, we are reticent to remove any parts and emphasize the importance of model testing and description so that the results are interpretable and trustworthy.

Reviewer 1 is unsure to which model configurations the efficient drainage was applied – this drainage was applied to both mass conserving models (linked-cavity and poro-elastic), but not the non-mass conserving, zeroth order "leaky bucket" model. In this sense the similarity in of results between the three hydrology setups demonstrates the lack of importance (for a Hudson Strait scale ice stream) of process details as important as mass-conserving lateral transport and efficient drainage for surge frequency (fig. 10 a), surge speed up (fig. 10 b), importance of effective pressure in the basal velocity calculation (rNeffFact, fig. 12). Even comparing the surge duration and sliding velocity/effective pressure distribution between poro-elastic (mass-conserving with efficient drainage) and leaky-bucket (non-mass conserving without efficient drainage) – which have the same pressure closure – does not yield appreciable difference. It is surprizing that these two process differences – the main process details – do not matter for the given context. The comparison between these two systems with similar pressure closure but differences in the main process details is not made clear enough in our submission and will receive more attention in the revised manuscript.

Current runs are underway to show the model output under both SHMIP scenarios A3 and A5 to compare the behaviour of the efficient system in BraHms to that of GlaDS and will appear in the final manuscript.

We will implement all changes suggested by Reviewer 1 unless otherwise stated. Below we respond to the specific comments and some of the more technical suggestions.

- Section 2: We will elaborate the description of the efficient drainage and explicitly state that it applies to both of the mass-transporting systems (poro-elastic and linked-cavity). We will also be including further justification of this choice of diagnostic (non-dynamical) efficient drainage model as Reviewer 2 has suggested.

- Section 3: As stated above, we are currently running an additional comparison to the GlaDS output for scenario A5.

- Discussion: The context for our conclusions is given in the title and caveats as to other applications where the subglacial hydrology process details might be important are given in the discussion (*e.g.* line 573). We will underline caveats and model limitations in the discussion of the revised manuscript.

- Line 19: We are not saying that the role of subglacial hydrology at time scales longer than a season is clear, but are making the more limited statement that subglacial hydrology's role at time scales longer than multi-decadal (if not less) is *not* clear. We will leave this sentence as is in the revision.

- line 358: $T_{bp}$ is the basal temperature as stated earlier in the paragraph (line 357). We will leave this as is in the revised manuscript.

30

---

## Author Comment (AC2)

**Response to Reviewer 2 of article entitled "Surging of a Hudson Strait Scale Ice Stream: Subglacial hydrology matters but the process details don't"**

Matthew Drew[1] and Lev Tarasov[1]

[1]Department of Physics & Physical Oceanography, Memorial University of Newfoundland, St John's, Canada

**Correspondence:** Matthew Drew (mdrew@mun.ca)

We would like to thank Reviewer 2 for their detailed consideration of the manuscript and we feel that many of their suggestions will ameliorate the final draft.

Reviewer 2 raises concern with the treatment of the efficient drainage system.

While dynamically modelling flux through an efficient system might be ideal, the CFL criterion would impose prohibitively long run times for our context. For example, given a lower bound water velocity of 1 m/s in the channel system as measured by Chandler et al. (2013) and our (coarse) resolution of 50 km, a time step of 0.00158 model years is required. Because our down gradient routing, drainage solver is not restricted by CFL, the time step depends only on the inefficient system for our setup. This typically lies in the range of 0.5 to 0.25 model years (lower bound time step of 0.0625 yr). Directly modelling the efficient system would increase BraHms runtime anywhere from 150 to >300 fold rendering simulation of millenial scale variability infeasible. Dynamical changes in flow through the efficient system occur on diurnal to seasonal time scales. Given the disparity in time scales (we are examining centennial to millenial time scales) it seems unlikely that dynamical changes in the efficient system (requiring a dynamic model) would be a significant control on the longer scale variability (though it is difficult to rule such a coupling out completely in a non-linear system, and we will make this caveat explicit in the revised text). We are, however, keen to see any evidence or mechanistic description of how such a coupling could significantly occur across such a wide difference in timescales.

Arguably the most important aspect of subglacial hydrology is the mass conserving lateral transport, an aspect which we show does not matter for (*e.g.* ) surge frequency and strength. We will add a shortened version of the above to the justification of the efficient drainage model design in the revised manuscript.

Regarding the length and density of the paper and the polish of the discussion, Reviewer 2 has made many helpful suggestions (as had reviewer 1) (i.e. in "Minor Issues") which will address this point and improve the discussion which we intend to incorporate. We intend to incorporate all of the suggestions made by Reviewer 2 except where specifically stated otherwise herein. Below we address a subset of the points for which a bit more discussion is useful.

– 220: We did not attempt this and took the comparison with GlaDs output for SHMIP scenario A3 as sufficient verification in this vein.

25   – 250: We will visit this for the revised manuscript. The same domain is used for both the spatial and temporal convergence test, which we will make explicit in the revision.

  – 344: The assumption here is more so that the hydraulic potential is controlled by the slope in ice surface elevation (Johnson and Fastook, 2002). Given a lower viscosity for water versus ice and the same potential gradient, the velocities for water should be faster than ice. There could be exceptions to this but they are the minority.

30   – 428: By "sieving" we simply mean discarding all runs that lie outside the given metric ranges (akin to history matching, explained in Tarasov and Goldstein (2023)). For example, in the case of the geometry sieve these metrics are the maximum ice thickness and north-south extent and those runs with ice which is too thick or thin or extents too large or small were excluded from the sieved set. This definition will be made clear in the revision We did not consider a formal Bayesian calibration but this would be interesting. However, such a calibration is outside the scope of this paper and the
35 numbers of runs making it through the sieves are sufficient for our statistical analysis without re-running the ensembles with updated priors.

  – 436: This non-parametric sensitivity method is novel to this study. We did consider other methods, for example those of Borgonovo (2007) and Pianosi and Wagener (2015), however for those methods to converge even larger ensembles would be needed. This sensitivity method is *relatively* cheap. We will make this clear in the revision

40   – 450: The sieve is a random sampling of the dummy variable as it was not used in the actual model. As such it would be unexpected for the sieved distribution to be identical to the unsieved. The KDFs drop off at the maximum and minimum of the range (instead of immediately going to zero) due to the bandwidth method used. We used the gaussian_kde method of the scipy.stats package (Virtanen et al., 2020) with the default Scott method for bandwidth determination. We suspect this explanation would not add to the revised manuscript.

45   – 459: The inflection point indicates the point of diminshing returns in parameter sensitivity with decreasing parameter rank (by sensitivity metric). The third degree polynomial fit between parameter rank and sensitivity metric is used to estimate this inflection in sensitivity. This will be explained in the manuscript.

  – 510: Because the surge duration is sensitive to the number of events occuring within a fixed time period, the duration signal is different at different frequencies. It is therefore necessary to assess differences in surge duration between model
50 setups across frequencies. This difference in duration is also a caveat to our central conclusion of the unimportance of process details for large ice stream surging. We will expand this in the revised manuscript.

**References**

Borgonovo, E.: A new uncertainty importance measure, Reliability Engineering & System Safety, 92, 771–784, https://doi.org/10.1016/j.ress.2006.04.015, 2007.

Chandler, D. M., Wadham, J. L., Lis, G. P., Cowton, T., Sole, A., Bartholomew, I., Telling, J., Nienow, P., Bagshaw, E. B., Mair, D., and et al.: Evolution of the subglacial drainage system beneath the Greenland Ice Sheet revealed by tracers, Nature Geoscience, 6, 195–198, https://doi.org/10.1038/ngeo1737, 2013.

Johnson, J. and Fastook, J. L.: Northern Hemisphere glaciation and its sensitivity to basal melt water, Quaternary International, 95–96, 65–74, https://doi.org/10.1016/s1040-6182(02)00028-9, 2002.

Pianosi, F. and Wagener, T.: A simple and efficient method for global sensitivity analysis based on cumulative distribution functions, Environmental Modelling & Software, 67, 1–11, https://doi.org/10.1016/j.envsoft.2015.01.004, 2015.

Tarasov, L. and Goldstein, M.: Assessing uncertainty in past ice and climate evolution: overview, stepping-stones, and challenges, https://doi.org/10.5194/egusphere-2022-1410, 2023.

Virtanen, P., Gommers, R., Oliphant, T. E., Haberland, M., Reddy, T., Cournapeau, D., Burovski, E., Peterson, P., Weckesser, W., Bright, J., van der Walt, S. J., Brett, M., Wilson, J., Millman, K. J., Mayorov, N., Nelson, A. R. J., Jones, E., Kern, R., Larson, E., Carey, C. J., Polat, İ., Feng, Y., Moore, E. W., VanderPlas, J., Laxalde, D., Perktold, J., Cimrman, R., Henriksen, I., Quintero, E. A., Harris, C. R., Archibald, A. M., Ribeiro, A. H., Pedregosa, F., van Mulbregt, P., and SciPy 1.0 Contributors: SciPy 1.0: Fundamental Algorithms for Scientific Computing in Python, Nature Methods, 17, 261–272, https://doi.org/10.1038/s41592-019-0686-2, 2020.

55

60

65

---

## Author Response (AR1)

**Updates following the suggestions of reviewer 2**

Matthew Drew[1] and Lev Tarasov[1]

[1]Department of Physics & Physical Oceanography, Memorial University of Newfoundland, St John's, Canada

**Correspondence:** Matthew Drew (mdrew@mun.ca)

We thank the anonymous reviewers for their generous treatment of the manuscript and helpful suggestions. Below we detail the changes made with the reviewer comments (latter shown in **bold**).

**1 Response to Reviewer 1**

**1.1 General comments by Reviewer 1**

5 **This study investigates the implications of introducing different subglacial hydrology model formulation for the surge dynamics of an Hudson Strait like ice stream. The manuscript first introduces the different approaches that are used for the subglacial hydrology modelling before providing a verification of theses implementations. The authors then present a framework for long term simulations of an Hudson Strait like experimental design that allows to perform a large ensemble of simulations on long time scales. The results of the ensemble is then presented both to emphasize the**
10 **parameter sensitivity of the different hydrology model and the differences that arise between the different approaches.**

**I found this study to be overall well presented and clear on an interesting and topic that as not been studied in details before. In an effort to give a clear description of their approach and models, the authors end up with a quite lengthy paper but I expect that it can not really be avoided here, I only noted a few places where some paragraphs may be omitted which would make room for more critical information.**

15 **As a general note, I am under the impression that some of the conclusion of the study and particularly the one relative to the unimportance of "process details" is overstated. I agree that the results shown here on the Hudson Strait surges frequencies seems to be unchanged by the different hydrology modelling approach used here but it feels to me that it is a bit thin to have such a strong conclusion.**

**Regarding the hydrology processes themselves, I am unsure if the efficient drainage is applied to the linked cavity**
20 **system only or two both the mass transporting hydrology setups. I also am not sure why this part of the model is not compared to the GlaDS model, to me the shift between efficient and inefficient drainage is one of the crucial point of a subglacial hydrology model and I would be very interested to see how the approaches used here compare to GlaDS with a higher recharge scenario (like A5 from SHMIP).**

**The presentation of the ensemble results and the sieve treatment made this part easy to read and interpret and that**
25 **was a very interesting read, I noted a few places where referencing could be improved. I would also like to suggest a few changes to the figures that are listed in the minor comments bellow and would improve readability.**

**Author response to general comments**

We have implemented nearly all of the helpful suggestions from Reviewer 1 The details of the various model configurations are now more clearly presented in the revised manuscript (e.g. table 2). We hope this will help readers draw the same conclusions as those described in the manuscript. In particular this will highlight that despite significant differences in process inclusion (such as lateral transport or efficient drainage), similar surge characteristics can be reproduced across those configurations which include subglacial hydrology. We have softened the conclusion in the title saying that the "process details mostly don't" matter. Additionally, the manuscript has been shortened by moving the model verification details to the supplement.

The efficient drainage was applied to both mass conserving models (linked-cavity and poro-elastic), but not the non-mass conserving, zeroth order "leaky bucket" model. The similar range in surge behaviour between the three hydrology setups demonstrates the lack of importance (for a Hudson Strait scale ice stream) of process details as important as mass-conserving lateral transport and efficient drainage for surge frequency (fig. 7 a) and surge speed up (fig. 7 b). Even comparing the surge duration and sliding velocity/effective pressure distribution between poro-elastic (mass-conserving with efficient drainage) and leaky-bucket (non-mass conserving without efficient drainage) – which have the same pressure determination – does not yield appreciable difference given parametric uncertainties. All of this supports the conclusion that the process details are not important, though as stated above the title has been changed to reflect surge characteristics where the process details do have an influence, for example surge duration.

The efficient drainage system has not been compared to that of GlaDS because of a significant disparity in spatial and time scales targeted by the model (ice sheet/millennial scales versus glacial/annual scales) and the ensuing differences in process representation. The efficient drainage in GlaDS is a dynamically calculated R-channel drainage (Werder et al., 2013) whereas the efficient drainage in BrAHMs2.0 is diagnostic – routing water within a "very short" time frame relative to the centennial and longer time scales considered. A such the efficient drainage in BrAHMS2.0 does not calculate a flux and does not produce an effective pressure comparable to GlaDS under the steady conditions of (e.g.) SHMIP scenario A5.

There are a few other items which have not been changed. The reference to Hank et al. (2023) has been left in as this paper is now accepted in GMD and so the details on the basal temperature ramp can been examined there. The list of parameters comprising the parameter range justification section has been left as is so that the reader may interpret the parametric uncertainty evaluated in this manuscript. These and other minor points which have not been changed are described in detail below.

**1.2 Specific Comments**

– Introduction: **The reference to Hank et al. here is a bit problematic as it is not clear to me what we are talking about, perhaps a succinct description of the method developed in the Hank paper should be introduced here.**

The Hank et al. (2023) paper referenced here is now available in pre-print (and is now accepted for GMD) and those details can be seen there.

– Section 2: **I find the description of the inefficient drainage systems to be well presented here but I am less convinced by the description for the efficient drainage system, I am not sure for example if the efficient drainage applies to both poro-elastic and linked cavity systems or to only one of those.**

The details of the various model configurations applied to the experiments herein are now shown in table 2 for clarity. Furthermore, we've added detailed arguments (with one caveat due to non-linearity) as to why the diagnostic treatment of efficient drainage is reasonable (and necessary for stated contexts): " Fully modelling the process of efficient drainage of water through the channel system would require very short time steps due to CFL (Courant et al., 1928) restrictions and consequently prohibitively long run times. Given the disparity in time scale between efficient drainage (sub-annual) and the dynamical behaviour examined here (centennial to millennial scale surging), it is unlikely that dynamically versus diagnostically modelling the efficient drainage will have an effect on the longer time scale surging, though as the model is non-linear there is potential for propagation across time scales."

– Section 3: **The sub-sectioning of this part could be improved, perhaps the very short introduc- tion can be replaced by the current sub-section 3.2 which would serve the purpose of introduction here. I would also suggest to have a specific section for the hydrology model verification which in my opinion is an important part of the study. The assumption of the hydrology model are also quite important in the description of the model and should in my opinion appear in the section 3 rather than in the sup- plementary. In order to shorten the main manuscript, section 3.1.1 could be moved to the supplementary materials. Regarding the subsection on model validation, I am unsure of the necessity of the introductory paragraph I would however like to see a comparison of the model to GlaDS with a higher input scenario to assess the changes that are due to this part of the model.**

We have moved section 3.2 to the introduction of this section and added a "Basal Drag Coupling" subsection (now 3.2) to clarify the model description. Also a new table 2 details the various model configurations. Section 3.1.1 has been moved to the appendices (C). The model verification details have also been moved to the appendices (D) as we suspect most readers will be less interested in those details. The detailed discussion of model assumptions has been left there as well for the same reason while the assumptions are summarized in the relevant model description sections.

Including an additional SHMIP scenario (A5 de Fleurian et al., 2018) in this appendix to compare this model against GlaDS (Werder et al., 2013) was not possible as the efficient drainage model here is significantly different from that in GlaDS: it is either switched on or off giving step changes in effective pressure at short time scales and does not calculate a flux. Results from this efficient drainage model are not interpretable in the short scales considered in SHMIP.

– Section 4: **Part of this section might be reduced, I feel that the parameters for which only a few lines are given with no strong argumentation could only appear in a table with a reference cited to justify the range. I am not sure of the temperature description that is given in equation 18. Following the equation given there and the map of LISsq the temperature at the northern end of the domain would not be $T_{north}$ but $T_{north} + 5000 T_{grad}$ also, the elevation dependant term is summed with the others but the lapse rate is positive which would give higher temperature at higher elevation.**

Though the justification of some parameters is only a few lines we feel it is important to lay out reasoning such that the results are interpretable. The equation (15) giving surface temperature has been fixed.

- Discussion: **As for the title I feel that the discussion here is a bit to strong in term of wording. I agree that the details of hydrology approaches presented here do not impact greatly the frequency of surges but they may be important for other applications (as shownin figure 13 for example). It feels also necessary here to underline the shortcomings of the present study and their potential implications on the conclusion.**

The conclusion as stated in the title has been softened with "mostly" and we have given suggestions as to when the details of subglacial hydrology may matter (e.g. the discussion of fig. 11 to which Reviewer 1 refers): " Though periodicity and strength of surges are similar between the three hydrology bearing experiments, an interesting distinction occurs when examining the duration of events at varied frequencies. The stabilizing negative feedback of increasing effective pressure at higher basal velocities in the linked-cavity pressure closure gives surge durations longer (up to double, depending on frequency) than those of the diagnostic pressure closure of the poro-elastic and leaky-bucket hydrologies. This feedback also results in a bimodal effective pressure distribution (i.e. fig. 11). When studying ice stream surge behaviour, any of the hydrologies may give the same surge response in terms of frequency and strength of surges. If the study requires a more granular understanding of how long the surge was active, for example when studying the surge timing of multiple ice streams in a catchment (e.g. Payne, 1998; Anandakrishnan and Alley, 1997) or the lifespan of palaeo-ice streams, our results suggest that accounting for the appropriate hydrology system is required."

We have also added several shortcomings in the Discussion section:

- a caveat that in non-linear systems effects may propagate across time scales – "However, in a non-linear system, such a control across scales cannot be fully ruled out."

- a caveat that while the basal sliding rule was parameterized to probe sensitivities there, a full probe of additional forms was not performed: "Future work on the contribution of the subglacial hydrologic system could build on this work by further examining the role of the form of the basal sliding rule. We have partially captured this contribution through the parameterized exponentiation of the the stress term, sliding coefficient, and effective pressure normalization. However additional forms could be probed – for example other exponents on the effective pressure term or a coulomb plastic relationship for sliding based on soft sediment deformation (potentially impactful in the soft bedded synthetic Hudson Strait)."

- a caveat that while comprehensive, the study lacks application to more realistic systems and observational constraint: "A larger limitation of this study is its synthetic, process focus. Ideally these systems would be applied to more realistic boundary conditions for which real constraint data might exist, for example velocity increase and meltwater discharge within a catchment for a large contemporary ice stream."

**2 Response to Technical Comments**

- **line 7: Is "BraHms" here an acronym, if yes it should be spelled.**

125    fixed

- **line 8: I am not sure what "systems" are referred to here.**

Replaced topologies with systems to clarify what is meant by systems later on.

- **Line 19: I would not say that the role of subglacial hydrology at time scales larger than a season is clear, so perhaps this sentence can be rephrased.**

130    Left as is. As stated in the response to review, we are not saying that the role of subglacial hydrology at time scales longer than a season is clear, but are making the more limited statement that subglacial hydrology's role at time scales longer than multi-decadal (if not less) is *not* clear

- **line 21: The sentence ending on this line is not the clearest and could be reworded.**

removed this sentence as it was unclear and did not add value

135    - **line 32: Studies by Fürst et al. (2015); Shannon et al. (2013) might be relevant here and could be added.**

It is not clear how these references are relevant

- **line 43: "begin to rapidly slide" "to" is missing.**

reworded

- **line 48: References for the hydrology formulations should be added.**

140    refs added

- **line 52: A reference is missing for GSM.**

added Tarasov et al. (2012) and Tarasov and Peltier (2007).

- **line 64: I expect that "HS" stands for Hudson Strait but it has not been introduced before.**

fixed

145    - **line 70: It should be "or" in place of "and".**

fixed

- **line 78: I am not sure why porous media is used here where poro-elastic is used everywhere else.**

fixed

- line 79: **Flowers (2015) should be added here.**

  added

- line 91: **Smith et al. (2021) could be added here.**

  Reference not added – relevance unclear

- line 94: **I find this formulation unclear.**

  clarified, changed to: "The contrast between the order kilometre or larger model scales and the order metre or smaller process scales permits inefficient flow"

- line 116: **I feel that the formulation here could be shortened giving only the new equations are the exponents of equation 1.**

  removed

- line 121: **Pwater is already defined above.**

  removed

- line 133: **I feel that the formulation here could be shortened giving only the new equations are the exponents of equation 1.**

  shortened

- line 133: **k is referred to here bu K is in the equation.**

  fixed

- line 135: **Nef f is usually referred to as effective pressure, is there a reason for it being pressure closure here?**

  Changed to "effective pressure"

- line 141: **Nef f was defined above already.**

  removed

- line 146: **There is a missing reference here.**

  fixed

- table 1: **The acronyms of the table should be described in the caption.**

  fixed

- line 174: **Numbering is missing for those equations and it should be hcav in place of h.**

  fixed

– Equation 9: **phieng and mt are not described.**

fixed

– line 182: **drain and smelt could be added on this line for description.**

fixed

– line 183: **A comma is missing in the reference.**

fixed

– line 198: **"v2" is missing for BraHms.**

fixed

– line 245: **The sentence starting on this line should be rephrased.**

fixed

– line 258: **Figure 1 shows the convergence for all parameters not only Nef f so I am not sure here what Lambda stands for.**

fixed

– line 307: **"Bay/Strait" is capitalised here but not bellow.**

fixed

– line 307: **"HEINO" here needs a reference.**

added

– Table 2: **This table is missing from my version of the manuscript.**

fixed

– equation 20: **There is no description of the parameters of this equation.**

fixed

– line 350: **kmin value should be fixed.**

unsure what is meant here – $k_{min}$ is set by two model parameters, kmax and kratio

– line 350: **Is there a reason to have upper case and lower case "k" for conductivities.**

made consistent

– figure 5: **Colours on this figure should be changed for more colourblind friendly versions.**

These histograms have been changed to line plots and linestyles changed to make them accessible to a broader audience.

- figure 5: **I am not sure which Flow speed is referred to on the x axis.**

  added "basal water flow speed" in caption

- equation 21: **Operator log and arctan should not be slanted fonts.**

  fixed

- equation 21: **is there a reason for the renaming of hc to h_crit_dpe here?**

  fixed

- line 358: **I have not seen Tbp used anywhere else.**

  defined

- equation 22: **I have not seen Kf defined anywhere.**

  defined

- equation 22: **Tf roz here and in the text have different font.**

  font adjusted

- line 371: **"e.g." should be before the reference.**

  fixed

- line 372: **I am not sure that the precision on the referencing are needed here.**

  the precision here is intended to highlight the lack of justification for these parameter values in a traceable way

- line 385: **I am not sure of the meaning of "communition".**

  spelling fixed, sentence adjusted

- line 390: **See my comment above on the clarification for the Hank paper.**

  The pre-print detailing this reasoning/testing from Hank et al. (2023) is now available

- line 397: **I am not sure of the point of the sentence starting on this line.**

  This has been adjusted to " $Q_{scale}$ is a scale factor adjusting for subgrid uncertainty – small scale fluctuations in flux may trigger a run-away tunnelling positive feedback affecting the larger scale."

- line 402: **I think that the reference to fig. 6 here is misplaced.**

  fixed

- figure 6a: **Label is missing on the y axis.**

  label added to vertical axis

230     – line 452: **Shouldn't reference here be to figure 8 rather than 12.**

       fixed

    – line 457: **Shouldn't reference here be to figure 8 rather than 12.**

       fixed

    – figure 8: **The horizontal line is not described in the caption.**

235       The horizontal line is now described

    – figure 9: **Line colours should be given in the caption, the legend should be moved for clarity and upper case letter used as in the caption.**

       Plot elements defined in caption

    – line 476: **Reference to figure 9 should be added here.**

240       added

    – line 489: **Reference to figure 9 should be added here.**

       added

    – line 497: **I am not sure what "tbl. 3" references to is it in the referenced study?**

       fixed

245     – line 504: **"on of HSIS" either "on" or "of " should be removed.**

       removed

    – figure 10: **The colours between PE and NH should be changed for colourblind friendliness.**

       These figures have been changed to line plots for legibility and line styles have been added for accessibility to a broader audience.

250     – figure 10: **On this type of figures perhaps a line would be clearer than the histogram to compare the different models.**

       changed to line plots

    – figure 10: **"Run Density" of panel a is missing ticks and labels.**

       vertical axes tick marks added

255     – line 527: **"Fig 12" should be referenced here.**

       added

– figure 11: **The colours between PE and NH should be changed for colourblind friendliness.**

different marker symbols have been added for colourblind friendliness

– figure 11: **Upper panel y-axis of panel a is missing a label.**

The upper panel of part a of this figure has been removed as it did not add much

– line 535: **A reference to figure 13 should be added here.**

added

– figure 12: **The horizontal line is not described in the caption.**

added

– figure 12: **I wonder why the delineation between sensitive and insensitive is not presented as in Figure 8.**

clarified in figure caption: "Inflection is weak in each case and so is not used to delineate between sensitive and insensitive parameters"

– figure 13: **The font on this figure is a bit small.**

font size has been adjusted

– line 551: **The paragraph starting on this line could be clearer.**

This paragraph in question:

"That the range of HS surge behaviour (frequency, amplitude) cannot be realistically captured absent subglacial hydrology is not due to a faster sliding coefficient. The effective pressure normalization is more than balanced by the effective pressure in the denominator in the basal velocity calculation. This is shown by the slower velocity distributions (min, mode, max) in the hydrology ensembles relative to the no hydrology configuration in fig. . The range of soft bed sliding coefficient covered in each ensemble approaches the bounds of plausibility – rmu $\in [0.01, 4.0]$, where rmu is scaled to give a 3 km/a sliding velocity for 30 kPa basal drag. Increasing the sliding coefficient does not capture HS surge behaviour"

has been changed to:

" Plausibly, one might expect that simply increasing the sliding coefficient in the no hydrology case would generate more surges. We therefore compared the basal velocity distributions between the configurations. The velocity distributions (min, mode, max) in the hydrology ensembles were slower relative to the no hydrology configuration in fig.The range of soft bed sliding coefficient covered in each ensemble approaches the bounds of plausibility – $c_{rmu} \in [0.01, 4.0]$, where $c_{rmu}$ is scaled to give a 3 km/a sliding velocity for 30 kPa basal drag. HS surge behaviour cannot be captured by increasing the sliding coefficient."

- line 554: **The number of the referenced figure is missing.**

  figure number has been fixed

- figure A1 and A3: **Same colour issue as before, and the grey region is not described in the caption, ticks and labels on the left panel are missing.**

  These have been adjusted as with previous similar figures

- figure A2 and A4: **Same issues as figure 11.**

  These have been adjusted as with previous similar figures

- figure C1: **The colours should be changed for colourblind friendliness.**

  The colours in this figure have been replaced with varied line styles

- line 634: **The number of the equation referenced here is missing.**

  The equation number has been fixed

**3 Anonymous Reviewer 2**

**3.1 Major comments by Reviewer 2**

**"Surging of a Hudson Strait Scale Ice Stream: Subglacial hydrology matters but the process details don't" by Drew and Tarasov presents a subglacial hydrology model implemented in an ice-sheet model geared towards paleo applications. The manuscript describes three versions of the model, a series of verification tests, and the application to a simplified Laurentide Ice Sheet configuration. The authors perform large ensembles with each version of the model to assess parameter sensitivity of the model for applications to ice-sheet thickness and surging. They find that the inclusion of subglacial hydrology improves the surging behaviour of the model, but the choice of form of subglacial hydrology among the three forms considered is not critical for primary quantities of interest.**

**The study is impressive in its thoroughness. The comparison of the different styles of subglacial hydrology models for the application of interest is exhaustive and highlights similarities and differences that are often left to speculation. I also commend the authors for the detailed model verification they perform, which is important but often skipped. The size of the ensembles performed is truly impressive and a great achievement. The size of the parameter space explored provides a lot of confidence on the conclusions. One longstanding problem in subglacial hydrology modeling is not knowing which models to use or what parameter values to use because of the large uncertainty in both choices. The large ensembles described here are one of the few detailed assessments of this issue that I've seen.**

**Despite these strengths, the study suffers from two major issues. First, the cursory consideration of channelized/efficient drainage makes it hard to generalize the results to realistic conditions. Second, the manuscript is long, dense, and feels unfinished. I elaborate on these issues below.**

**Major Issues**

**Inadequate consideration of efficient/channelized drainage. The authors admit they have implemented a somewhat ad hoc solution to efficient drainage, which is fine, particularly given the long time scale and relatively coarse resolution modeling they are targeting. However, this choice severely limits their ability to address some of the main questions in the manuscript. For example, on line 33 is stated the major question of "And if so, to what extent are the structural details of the hydrological system important for this context, especially given the rest of the system uncertainties?" Similarly, the somewhat provocative title implies the full range of process details have been considered, when arguably the most important process detail has not. I could see two ways to address this issue: revise the model, the simulations, and the manuscript to properly consider efficient drainage, or modify the presentation to more carefully acknowledge this limitation. Obviously, the former is unlikely to be practical. For example, in addition to the examples listed above, line 47 avoids mentioning efficient drainage at all.**

**Manuscript is complicated and feels unfinished. The Discussion section is hard to follow and missing many figure references. Additionally, there is no discussion of model limitations (see item 1) or comparison of results to the existing literature. For how long the manuscript is, the conclusion is quite rushed. Specific examples are listed below.**

**Author response to Reviewer 2 major comments**

Reviewer 2 raises concern with the treatment of the efficient drainage system. While dynamically modelling flux through an efficient system might be ideal, the CFL criterion would impose prohibitively long run times for our context. For example, given a lower bound water velocity of 1 m/s in the channel system as measured by Chandler et al. (2013) and our (coarse) resolution of 50 km, a time step of 0.00158 model years is required. Because our down gradient routing, drainage solver is not restricted by CFL, the time step depends only on the inefficient system for our setup. This typically lies in the range of 0.5 to 0.25 model years (lower bound time step of 0.0625 yr). Directly modelling the efficient system would increase BraHms runtime anywhere from 150 to >300 fold rendering simulation of millennial scale variability infeasible. Dynamical changes in flow through the efficient system occur on diurnal to seasonal time scales. Given the disparity in time scales (we are examining centennial to millennial time scales) it seems unlikely that dynamical changes in the efficient system (requiring a dynamic model) would be a significant control on the longer scale variability (though it is difficult to rule such a coupling out completely in a non-linear system, and we now make this caveat explicit in the revised text). We are, however, keen to see any evidence or mechanistic description of how such a coupling could significantly occur across such a wide difference in timescales.

Arguably the most important aspect of subglacial hydrology is the mass conserving lateral transport, an aspect which we show does not matter for (e.g.) surge frequency and strength. We have added a shortened version of the above to the justification of the efficient drainage model design in the revised manuscript and discussed it further in the discussion section:

" Given the disparity in time scale between efficient drainage (sub-annual) and the dynamical behaviour examined here (centennial to millennial scale surging), it is unlikely that dynamically versus diagnostically modelling the efficient drainage will have an effect on the longer time scale surging."

Further, the efficient drainage coupling is not applied to the leaky-bucket configuration which produces the same range of surge characteristics as the linked-cavity and poro-elastic configurations which do couple in the efficient drainage. The

down gradient routing scheme used to approximate the channel system effectively overestimates how quickly an efficient drainage system can modify the hydraulic state and thereby effective pressure of an ice sheet. That the surging behaviour can be reproduced across all three setups indicates that the efficient drainage is not necessary for reproducing the range of surging characteristics examined in this manuscript. We have added these points to the discussion section in the revised manuscript.

Regarding the length and density of the paper and the polish of the discussion, Reviewer 2 has made many helpful suggestions (as had reviewer 1) (i.e. in "Minor Issues") which will address this point and improve the discussion which we intend to incorporate. We intend to incorporate all of the suggestions made by Reviewer 2 except where specifically stated otherwise herein. Below we address a subset of the points for which a bit more discussion is useful. Further, the model verification section has been moved to the supplement as this will be of interest to a narrower audience.

**3.2 Response to Minor Issues**

– 16: **If there is space in the abstract work limit, concisely state the feedbacks that the linked-cavity model has that the other don't.**

Added "The linked-cavity and poro-elastic configurations include an efficient drainage scheme while the leaky-bucket does not. All three systems have a positive feedback on velocity whereby faster basal velocities increase melt supply. The poro-elastic and leaky-bucket systems have diagnostic effective pressure relationships – only the linked-cavity system has an additional negative feedback whereby faster velocities increase the dynamical effective pressure due to higher cavity opening rates. We examine the contribution of mass transport, efficient drainage, and the linked-cavity negative feedback to surging."

– 20: **"leading *to*"**

removed the sentence referred to by this comment

– 24: **Consider adding some more recent references as examples here (e.g. work in west Greenland about seasonal and diurnal variations).**

added reference to (Cook et al., 2021)

– 28: **I'm not sure "topology" is the right word here and throughout the manuscript. topology is defined as "the way in which constituent parts are interrelated or arranged". I think what you may mean is system form or structure.**

the word "topology" has been replaced by "system"

– 29: **You could be as far as to say these issues limit application of subglacial hydrology models generally.**

changed " These uncertainties hinder widespread adoption of subglacial hydrology models in glacial cycle scale ice sheet modelling (Flowers, 2018)" to " These uncertainties hinder widespread adoption of subglacial hydrology models in earth systems models in general and glacial cycle scale ice sheet modelling in particular (Flowers, 2018)"

- 39: **Also consider the mechanisms of (Bassis et al., 2017).**

Ocean forcing of ice dynamics is outside the scope of ice sheet internal oscillations examined in this paper. The reference to Bassis et al. (2017) has not been added.

- 49-51: **This detail would be more appropriate in the Methods than the Introduction.**

Moved to " The transition from frozen to temperate is smoothed to more realistically capture the transition to sliding (as in model (d) of Calov et al. (2010)) following the work of Hank et al. (2023). " in the Model Description section.

- 56-59: **These details are also distracting in the Introduction.**

We have left this as is as these details provide important introductory context for the scope of the experiments and conclusions.

- 64: **quantify what you mean by "large". These ensembles are impressively large for ice-sheet model standards.**

the range of ensemble sizes has been added: "between 11 and 20 thousand members each"

- 72: **In terms of the two modes evolving to each other, it is important to mention the hysteresis in evolution (Schoof, 2010).**

Added text mentioning the hysteresis: "When the efficient system flux falls through different lower flux threshold, the system transitions to inefficient drainage, resulting in hysteresis Schoof (2010)."

- 84: **Include the citation for this statement. (Flowers and Clarke, 2002)**

included a reference to Flowers (2000): "The pressure closure has no theoretical basis and is based on a power law with empirically constrained parameters (Flowers, 2000)."

- 93: **Add citations, e.g. (Rada and Schoof, 2018)**

included a reference to Rada and Schoof (2018): "However, cavities can only drain water once they grow enough to join and form a connected network (Rada and Schoof, 2018). "

- 113: **The mass conservation equation should be moved to section 2.1. As presented, it implies that it is unique to the poro-elastic model, but it also applies to the linked-cavity model.**

This equation moved to the parent subsection (2.1) "Water sheet thickness is a continuum property used to describe the average amount of water in a grid cell. Changes in water thickness is given by the fluxes and the aggregate of sources and sinks, $m$, in the water transport eqn.

$$\frac{\partial h_{wb}}{\partial t} + \boldsymbol{\nabla} \cdot \boldsymbol{Q} = m \tag{1}$$

cell, $\boldsymbol{Q}$ is the subglacial water flux and $m$ is the aggregate of sources and sinks."

- 119: **Again, add reference for this equation (Flowers and Clarke, 2002)**

added a reference to (Flowers, 2000): "Water pressure in the elastic pore-space is set by eqn.(Flowers, 2000)"

- 146: **missing reference**

reference to Röthlisberger (1972) is fixed

- 165: **spelling: Arakawa. Also provide a reference for this terminology (Arakawa and Lamb 1977)**

Spelling fixed and reference added: "This model uses a finite volume discretization with a staggered Arakawa C grid (fluxes at interfaces Arakawa and Lamb, 1977)."

- 168: **"modified version of Schoof (2010) as in that of Werder et al. (2013) and Bueler and van Pelt (2015)" It is unclear what is meant by this – each of those papers are very dense.**

the modification to the cavity opening rate of Schoof (2010) has been added in the revised text: "modified version of Schoof (2010) as in that of Werder et al. (2013) and Bueler and van Pelt (2015) whereby the cavity opening rate is proportional to the difference in bed roughness and subglacial water sheet thickness."

- 175: **formatting problem around "e.g."**

formatting has been fixed – exempli gratia now written as e.g. instead of italicized

- eq 9: **replace curly braces with parentheses**

curly braces replaced with square:

$$\frac{\partial P_w}{\partial t} = \frac{\rho_w g}{\phi_{eng}} \left( -\boldsymbol{\nabla} \cdot \boldsymbol{Q} + m_t - u_b \left( h_r - h_{wb} \right) / l_r + c_2 [P_{ice} - P_w]^n \right)$$

- 181: **add comma after "surging"**

comma added after preposition: "In order to assess the importance of transport vs pressure determination in surging,"

- 194: **This statement seems to be a repeat of line 191.**

removed the sentence "scheme of Tarasov and Peltier (2006) (*i.e.*, with a modified hydraulic gradient). As subglacial water is routed down hydraulic gradient, minima in hydraulic potential are filled and excess water carried further down gradient until either no water remains or the drainage chain reaches the margin and the remaining water leaves the system. "

- 220: **Did you consider the semi-analytic solution presented in (Bueler and van Pelt, 2015)?**

We did not consider this solution

435      – 250: **Did you attempt to assess if the rate of convergence matches the numerical order of the discretization? (same for spatial resolution in sec. 3.3.3)**

We examined the rate of convergence in the model verification sections (now in supplement) and found the rate of convergence roughly matches the numerical order of the discretizations within the range of resolutions examined here and with the added caveat that these solutions are non-linear. This has been added in the supplement: "The rate of convergence is approximately quadratic as expected for the $\mathcal{O}\left(\Delta t^2\right)$ leap-frog trapezoidal scheme" and "The numerical order of the upwind and finite volume schemes used here is $\mathcal{O}\left(\Delta x\right)$. The approximately linear rate of convergence in fig.matches this numerical order."

– **Why were different domains used for the temporal and spatial resolution convergence tests?**

Different domains were not used, this has been fixed in the text.

– Fig. 1: **What are the other variables than effective pressure? Only effective pressure is mentioned in the text on line 259.**

The other variables have been removed, all outputs are now subglacial hydrologic outputs (effective pressure and basal water thickness). The x-axis has been rescaled to show the time step/grid cell size and reflect the convergence rate.

– 289: **missing closing parenthesis.**

added parenthesis in section moved to supplement: "(and uses finite volume discretization with a regular Cartesian grid)."

– 300: **Please state the model resolution. Also mention the model time step used somewhere in section 4.**

added grid resolution: "The simple climate allows a free southern margin determined by the background temperature and feedbacks giving a dynamically determined ice sheet geometry at 50 km horizontal resolution."

– 335: **Other references to consider for conductivity ranges: (Hewitt, 2011; Hager et al., 2022)**

Added references to Hager et al. (2022); Werder et al. (2013) for the choice of hydraulic conductivity range in section 4.5: "This range encapsulates values suggested by Hager et al. (2022) and Werder et al. (2013)."

– 344: **Please clarify how ice velocity provides a lower bound on water flow speed. Is the assumption that water moves with the ice and shares the same speed?**

Added a justification for the statement "Fast ice velocities (e.g. ≈1 km/a) give a loose lower bound on water flow speeds" based on large viscosity differences and small potential gradient differences: "Whereas the viscosities differ by many orders of magnitude ($10^{14}$Pa*s for ice versus $10^{-3}$Pa*s for water, Cuffey and Paterson, 2010), the pressure gradient forces are less dissimilar."

– 348: **Are you applying a flow law here? If so, state what it is.**

The applied flow law is now stated in the revised text: " Darcy-Weisbach flow speeds between $3 \times 10^{-4}$ and $1 \times 10^0$ m/s"

– 351: **Reword this sentence to make it clear that this equation is what Flowers et al. did, not a certain relationship. Also, it is not clear how you are using this relationship.**

Reworded and moved to Subglacial Hydrology Model subsection: " While in the linked-cavity model the hydraulic conductivity is a single parameter, in the poro-elastic model Flowers (2000) uses a meltwater thickness dependent arctan function for hydraulic conductivity to capture a transition from low to high permeability during expansion of the pore-space"

– 356-363: **This methodological detail should be moved to the model description.**

These details have been moved to Subglacial Hydrology Model subsection

– 395: **Is this the equation from Schoof? If so, please add the citation here as well.**

added the reference to Schoof (2010) in the Tunnel Switching Scalar subsection: "The flux threshold switch from inefficient to efficient drainage is given by the ratio of cavity opening due to sliding versus wall melting from viscous heating (Schoof, 2010)"

– 405-408: **There is no description of how these parameters are used in GSM. Please add that here or to the model description.**

These details have been added in the GSM description section

– 4.12: **Please divide this section into a portion describing the ensemble design and a separate section describing the results.**

we will leave the details in this subsection on metric selection and method of calculation in the experimental design rather than moving them to the results section

– 416: **These are very large numbers of runs for ice-sheet modeling! Can you provide some details about how long a single run and/or entire ensemble takes to run and what computational resources were used?**

Details on runtime and computational resources have been added in the subsection Ensemble Design and Parameter Sensitivity: "Ensembles were run on a heterogeneous Linux cluster with 24-32 Gb RAM and 8-24 xeon or opteron cores per node, clock speeds ranging 2.4-2.7 GHz and a total of 652 cores. Runtimes averaged about 3 hours."

– 418: **Please add a table listing the parameters varied in each ensemble. The parameter sensitivity figures and discussion are extremely hard to follow without knowing what the parameters are.**

These parameter details are available in Table 1, a reference to this table has been added to the revised text: "the number of parameters in each setup (15 in LC, 16 in PE, 12 in LB and 9 in NH, shown in tbl."

– 428: **Is there a reference for this sieve technique? Regardless, can you provide a more explicit definition of what that means? Also, did you consider performing a formal Bayesian calibration to the desired ice-sheet thickness instead? It would seem you have large enough sample sizes to do that.**

By "sieving" we simply mean discarding all runs that lie outside the given metric ranges (akin to history matching, explained in Tarasov and Goldstein (2023)). For example, in the case of the geometry sieve these metrics are the maximum ice thickness and north-south extent and those runs with ice which is too thick or thin or extents too large or small were excluded from the sieved set. Tarasov and Goldstein (2023) describes what meaningful Bayesian inference entails, and we are far from that within the community or this study.

An explanation of sieving has been added to the revised text: "Here we study surge behaviour at scale similar to the Laurentide ice sheet by sieving (discarding runs outside the target metric ranges) the ensembles according to maximum ice sheet thickness and North-South extent."

– 436: **Similarly, is there a reference for the non-parametric sensitivity method you are employing here, or is it novel to the study? Why not using existing non-parametric sensitivity methods?**

This non-parametric sensitivity method is novel to this study. We did consider other methods, for example those of Borgonovo (2007) and Pianosi and Wagener (2015), however for those methods to converge even larger ensembles would be needed. This sensitivity method is *relatively* cheap. We have made this clear in the revision: " The importance of hydrology parameters to determining ice sheet geometry can be probed with sensitivity analysis. Local sensitivity analysis methods neglect interaction terms important for studying feedbacks in coupled models and so are not applicable here (Saltelli et al., 2008). Meanwhile variance based (Sobol, 2001) methods require assumptions about the sampling structure of the underlying inputs. The trouble with coupled models is they can be unstable, as such there are incomplete runs which render sampling structure assumptions moot. There are other non-parametric sensitivity methods which do not require assumptions about the input sample distribution but these require sample sizes even larger than those presented here in order to converge (e.g. Borgonovo, 2007; Pianosi and Wagener, 2015)" and " We develop a novel non-parametric method to measure sensitivity"

– 450: **Why is the dummy parameter not identical before and after sieving? Also, why do the unsieved KDFs in Figure 7 a and b drop off near the minimum and maximum values in the range?**

The sieve is a random sampling of the dummy variable as it was not used in the actual model. As such it would be unexpected for the sieved distribution to be identical to the unsieved. The KDFs drop off at the maximum and minimum of the range (instead of immediately going to zero) due to the bandwidth method used. We used the gaussian_kde method of the scipy.stats package (Virtanen et al., 2020) with the default Scott method for bandwidth determination. We think these details would not add to the revised manuscript so have no relevant changes there.

– **What is the difference between Figures 7 and 12? Why is Figure 12 introduced after Fig. 7 on line 452?**

Reviewer 2 is referring to Fig. 8 and 12 which use two different sieves: 8 shows the sensitivity results for an ice sheet geometry sieve ($\mathcal{S}^0_{geom}$ in the revised text) and 12 shows results for geometry and surging metrics sieve ($\mathcal{S}_{surge}$ in the revised text). These subsets have been detailed in a table (table 2) in the revised text to improve readability.

– 459: **Please justify why an inflection indicates which parameters could be fixed. Also, justify fitting a polynomial to categories of data. It does not seem that you do anything with the categories of sensitive and insensitive parameters, so this extra complexity might not be needed.**

The inflection point indicates the point of diminishing returns in parameter sensitivity with decreasing parameter rank (by sensitivity metric). The third degree polynomial fit between parameter rank and sensitivity metric is used to estimate this inflection in sensitivity. This has been explained in the revised manuscript: " This inflection point is an approximate indication of the diminishing sensitivities in the model setup. As such, parameters to the right of this point are taken as sensitive and those to the left are considered insensitive and could be fixed for the purposes of geometry."

– 460: **"The change in sensitivity metric from one parameter to the next highest is not strictly monotonic". They look monotonic. Are they sorted by increasing KDF diff? If so, by definition, they increase monotonically.**

These are monotonic and this has been fixed in the text

– 463: **Can you could identify which parameter are hydrology-related (and possibly other categories) in the figure using a colour or symbol? That would greatly help the utility of this figure in conveying which classes of parameters are most important (which could be discussed in 465-471).**

These colours have been added

– Figure 8: **Please make it explicit what the sieve criterion being evaluated in this figure is.**

The sieves used in fig 8 and 12 have been made explicit with references to the sieve table (table 2) added.

– Section 4.13: **This section would benefit from referring to some examples of what identified surges look like. It looks like that is what Figure 9 is, but it is never referenced in the text! Also, I recommend changing the section title to something like "Definition of surge metrics".**

the section title has been changed and the figure showing an example run and events picked has been referenced throughout the section

– 500: **The percentages listed are much smaller than 1/3-1/4.**

These fractions have been updated to be consistent with the proportions in the text

– 509: **"runs with \*surge\* events"**

"surge" added to the text: "The no hydrology case, however does differ from those three: the rate of runs with surge events is significantly lower and the frequency"

– Fig. 10: **You use the term "Heinrich event metrics" in the caption, but in the text you say "surge events". Please use a consistent terminology, because this is an ad hoc working definition, and it will reduce confusion. Also, I wonder if line plots would be easier to distinguish the 4 models – they are hard to differentiate as is.**

We have made the terminology around the surge events consistent. We have kept the the plots as histograms as it will be more readily apparent to the reader that these are discretized distributions.

– 510-521: **This text looks like it should belong in the following section (5.1). Or consider removing it. What is the purpose of analyzing the surge duration as a function of frequency? This part of the manuscript is hard to follow, and I question what value it adds. Unless there is clear use for this information, I recommend removing this section to reduce the complexity of an already complex paper.**

We have added additional justification for analyzing duration as a function of frequency. This analysis highlights one difference found between the three subglacial hydrology systems: "As the duration of surge events necessarily depends on the frequency of those events (having more events in a time period decreases the maximum possible duration of those events), we examine surge duration as a function of the number of events (as shown by the horizontal purple lines in fig."

– 524: **The parenthetical statement should not be parenthetical, as it is critical for understanding what was done. Also, please reword it (e.g. three and twelve what?).**

parentheses removed, statement reworded

– 525: **"sensitivity ranking" – Is this referring to Figure 12? I'm feeling very lost in this section.**

references to figures and tables were added to clarify this section

– 527-533: **Again, colouring/marking the parameters by type is needed to make this figure interpretable.**

colouring of the points by parameter type has been added

– 535: **Is this referring to Figure 13? Also, is this the whole ensemble, or one of the sieved subsets? From Fig. 13 caption it looks like the entire ensemble. I'm not sure that is useful given the arbitrary nature of the parameter ranges considered. Why use the whole ensemble for this?**

a reference to figure 13 has been added and the use of full ensembles has been justified in the text

– 536: **"stark difference" Why? Please elaborate.**

added "This bifurcation of the effective pressures from a linked-cavity system show that it can sustain lower effective pressures than its poro-elastic and leaky-bucket counterparts" as elaboration.

– General: **I recommend a table defining and naming the various ensemble subsets (sieved results). The text can then refer to those subsets unambiguously.**

This table has been added (table 2)

– Figure 13: **How are both the whole ensemble and the warm-based portion of the ensembles shown in this one figure?**

language has been clarified in the revised text: " Two dimensional logarithmic histogram of effective pressure and velocity solutions for all warm based points across the parameter-space-time domain."

– 551-561: **This section is hard to follow and seems to be getting into specific details. Also, these results are not shown anywhere, making it harder to assess these statements. I suggest removing these paragraphs.**

This section has been clarified and references made to the supporting results

– 554: **Figure reference missing.**

Figure reference fixed

– 570: **This result may be a feature of the specific friction law used, which is fairly ad hoc. The paper is lacking a discussion section on model limitations and comparison to other studies. Please add such a section. For example, (Hewitt, 2013) compares poro-elastic and linked cavity formulations.**

The basal drag exponent for the soft bed sliding law used here is an ensemble parameter so that the sensitivity to the form of the basal drag law should be in good part captured. This is evident in the surge frequency sensitivity analysis – the soft bed sliding exponent is the second most sensitive parameter for the linked-cavity and poro-elastic hydrologies, behind the effective pressure normalization. However, we now explicitly acknowledge that this doesn't fully cover all plausible basal sliding laws : "FILL IN"

The details on the basal sliding law used have been added to section 3.2.1 in the revised text.

**References**

[revised manuscript text omitted]

---

## Author Response (AR2)

**Response to Editor on Minor Revisions**

Matthew Drew[1] and Lev Tarasov[1]

[1]Department of Physics & Physical Oceanography, Memorial University of Newfoundland, St John's, Canada

**Correspondence:** Matthew Drew (mdrew@mun.ca)

Thank you to the editor for their suggestions, all of which have been applied to the revised manuscript.